# Canonical normalizing flows for manifold learning

**Kyriakos Flouris**
Department of Information Technology
and Electrical Engineering
ETH Zürich
kflouris@vision.ee.ethz.ch

**Ender Konukoglu**
Department of Information Technology
and Electrical Engineering
ETH Zürich
kender@vision.ee.ethz.ch

## Abstract

Manifold learning flows are a class of generative modelling techniques that assume a low-dimensional manifold description of the data. The embedding of such a manifold into the high-dimensional space of the data is achieved via learnable invertible transformations. Therefore, once the manifold is properly aligned via a reconstruction loss, the probability density is tractable on the manifold and maximum likelihood can be used to optimize the network parameters. Naturally, the lower-dimensional representation of the data requires an injective-mapping. Recent approaches were able to enforce that the density aligns with the modelled manifold, while efficiently calculating the density volume-change term when embedding to the higher-dimensional space. However, unless the injective-mapping is analytically predefined, the learned manifold is not necessarily an *efficient representation* of the data. Namely, the latent dimensions of such models frequently learn an entangled intrinsic basis, with degenerate information being stored in each dimension. Alternatively, if a locally orthogonal and/or sparse basis is to be learned, here coined canonical intrinsic basis, it can serve in learning a more compact latent space representation. Toward this end, we propose a canonical manifold learning flow method, where a novel optimization objective enforces the transformation matrix to have few prominent and non-degenerate basis functions. We demonstrate that by minimizing the off-diagonal manifold metric elements $\ell_1$-norm, we can achieve such a basis, which is simultaneously sparse and/or orthogonal. Canonical manifold flow yields a more efficient use of the latent space, automatically generating fewer prominent and distinct dimensions to represent data, and consequently a better approximation of target distributions than other manifold flow methods in most experiments we conducted, resulting in lower FID scores. [1]

## 1 Introduction

Many emerging methods in generative modeling are based on the manifold hypothesis, i.e., higher-dimensional data are better described by a lower-dimensional sub-manifold embedded in an ambient space. These methods replace the bijective normalizing flow in the original construction [1–5] (NF) with an injective flow. NFs are constructed as a smooth bijective mapping, i.e. a homeomorphism, where the learned distribution density is supported on the set of dimension $D$ equal to the data dimension, where $\mathbb{R}^{\mathbb{D}}$ is the data space. In order to fulfill the manifold hypothesis, an emerging class of methods, namely manifold learning flow models (MLF) [6–9], model the latent distribution as a random variable in $\mathbb{R}^d$ where $d < D$, i.e. realize injective flows. Consequently, the density lives on some d-dimensional manifold $\mathcal{M}_d$ embedded in $\mathbb{R}^D$ as required. If $\mathcal{M}_d$ is known a-priori, [10, 11], the density transformation to $\mathbb{R}^D$ in MLF is a relatively straight forward affair. If the manifold is to be

---

[1]Code is available at https://github.com/k-flouris/cmf.

learned, the change-of-variable formula, i.e. the volume element, can be computationally inhibiting to calculate. Caterini et al [7] have recently solved this problem and established tractable likelihood maximization.

MLF methods have thus evolved to the point where the general manifold hypothesis is properly incorporated and the likelihood is efficiently calculated. However, it has been observed that the learned manifold can be suboptimal; for example, due to a nonuniform magnification factor [12] and the inability to learn complex topologies [13]. Furthermore, pushforward models such as MLFs, i.e., $d < D$, are designed to avoid over-fitting, but $d$ in $\mathbb{R}^d$ is still arbitrary unless determined by some meta-analysis method. As well as, there is often degeneracy in the information stored in each of these latent dimensions $d$. For instance, as we show in Section 4.1, consider modeling a two-dimension line with considerable random noise, i.e. a 'fuzzy line'. If $d = 2$, both latent dimensions will attempt to encompass the entire manifold in a degenerative use of latent dimensions. Particularly, this behavior of storing degenerate representations is inefficient and can lead to over-fitting, topological, and general learning pathologies. Alternatively, we postulate that motivating sparsity [14] and local orthogonality during learning can enhance the manifold learning process of MLFs.

To this end, we propose an optimization objective that either minimizes or finds orthogonal gradient attributions in the learned transformation of the MLF. Consequently, a more compact [15] and efficient latent space representation is obtained without degenerate information stored in different latent dimensions, in the same spirit as automatic relevance determination methods [16] or network regularization [17]. The relevant dimensions correspond to locally distinct tangent vectors on the data manifold, which can yield a natural ordering among them in the generations. Borrowing the term 'canonical basis', i.e., the set of linearly independent generalized eigenvectors of a matrix, we introduce Canonical manifold learning flows (CMF). In CMF, the normalizing flow is encouraged to learn a sparse and canonical intrinsic basis for the manifold. By utilizing the Jacobians calculated for likelihood maximization, the objective is achieved without additional computational cost.

## 2 Theoretical background

### 2.1 Normalizing and rectangular normalizing flows

First, we define a data space as $\mathcal{X}$ with samples $\{x_1, ..., x_n\}$, where $x_i \in \mathbb{R}^D$, and a latent space $\mathcal{Z} \subset \mathbb{R}^d$. If, $d = D$, then a normalizing flow is a $\phi$ parameterized diffeomorphism $q_\phi : \mathbb{R}^D \to \mathbb{R}^D$, i.e., a differentiable bijection with a differentiable inverse, that transforms samples from the latent space to the data space, $\mathcal{X} := q_\phi(\mathcal{Z})$. Optimization of $\phi$ is achieved via likelihood maximization, where due to the invertibility of $q_\phi$ the likelihood, $p_\mathcal{X}(x)$, can be calculated exactly from the latent probability distribution $p_\mathcal{Z}(z)$ using the transformation of distributions formula [18],

$$p_\mathcal{X}(x) = p_\mathcal{Z}(q_\phi^{-1}(x))|\det\mathbf{J}_{q_\phi}(q_\phi^{-1}(x))|^{-1}. \tag{1}$$

$\mathbf{J}$ denotes the Jacobian functional such that $\mathbf{J}_{q_\phi} = \partial x/\partial z$, i.e., the Jacobian of $q_\phi$ at $x = q_\phi(z)$. Here, $\mathcal{Z}$ is modelled by a random variable $z \sim p_\mathcal{Z}$ for a simple density $p_\mathcal{Z}$, e.g., a normal distribution. Therefore, for large $n$, a normalizing flow approximates well the distribution that gave rise to the samples in a dataset $\{x_i\}_{i=1}^n$ by transforming the samples $z \sim p_\mathcal{Z}$ with a trained $q_{\phi^*}$ if $\phi^* := \arg\max_\phi \sum_{i=1}^n \log p_\mathcal{X}(x_i)$ [4].

In the context of manifold learning flows, or rectangular normalizing flows, $d < D$ implies there is a lower dimensional manifold $\mathcal{M} \subset \mathbb{R}^d$ that is embedded into the $R^D$ with a smooth and injective transformation $g_\phi : \mathbb{R}^d \to \mathbb{R}^D$. In practice, $g_\phi = h_\eta \circ \mathrm{pad} \circ f_\theta$, where $h_\eta : \mathbb{R}^d \to \mathbb{R}^d$, $f_\theta : \mathbb{R}^D \to \mathbb{R}^D$ and $\mathrm{pad} : \mathbb{R}^d \to \mathbb{R}^D$. Padding can be used for trivial embedding, i.e., $\mathrm{pad}(\tilde{z}) = (\tilde{z}_0 \ldots \tilde{z}_{d-1}, 0 \ldots 0)$. The rectangular Jacobian of $\mathbf{J}_{g_\phi}$ is not invertible and thus, $\mathbf{J}_{g_\phi}^T \mathbf{J}_{g_\phi} \neq \mathbf{J}_{g_\phi}^2$, which implies that the volume element needs to be calculated fully, i.e., the square root of the determinant of $\mathbf{J}_{g_\phi}^T \mathbf{J}_{g_\phi}$ which is an equivalent expression for the general metric tensor in this context - see the appendix. This leads to a more generic formulation

$$p_\mathcal{X}(x) = p_\mathcal{Z}\left(g_\phi^{-1}(x)\right)|\det\mathbf{J}_{g_\phi}^T(g_\phi^{-1}(x))\mathbf{J}_{g_\phi}(g_\phi^{-1}(x))|^{-1/2}. \tag{2}$$

However, note that maximizing $p_\mathcal{X}(x)$ on its own would only maximize the likelihood of the projection of $x$ on the latent space, not ensuring that the model can reconstruct $x$ from $z$, i.e., the

modeled manifold may not be aligned with the observed data. To encourage $x \in \mathcal{M}$, namely to align the learned manifold to the data space, a reconstruction error needs to be minimized during optimization. The following loss is thus added to the maximum-likelihood loss function, as explained in [6]:

$$\sum_{i=1}^{n} \left\| x_i - g_\phi \left( g_\phi^{-1}(x) \right) \right\|_2^2. \tag{3}$$

Combining the logarithm of Equation (2) and using the hyperparameter $\beta > 0$ to adjust Equation (3) we arrive at the total Lagrangian to be maximized:

$$\phi^* = \text{argmax}_\phi \sum_{i=1}^{n} \left[ \log p_{\mathcal{Z}} \left( g_\phi^{-1}(x_i) \right) - \log |\det \mathbf{J}_{h_\eta}^T (g_\phi^{-1}(x_i))| \right.$$
$$\left. - \frac{1}{2} \log |\det \mathbf{J}_{f_\theta}^T (f_\theta^{-1}(x_i)) \mathbf{J}_{f_\theta} (f_\theta^{-1}(x_i))| - \beta \left\| x_i - g_\phi \left( g_\phi^{-1}(x_i) \right) \right\|_2^2 \right]. \tag{4}$$

### 2.2 Riemannian geometry and the metric tensor

A $d$ dimensional curved space is represented by a Riemannian manifold $\mathcal{M} \subset \mathbb{R}^d$, which is locally described by a smooth diffeomorphism $\mathbf{h}$, called the chart. The set of tangential vectors attached to each point $\mathbf{y}$ on the manifold is called the tangent space $T_{\mathbf{y}}\mathcal{M}$. All the vector quantities are represented as elements of $T_{\mathbf{y}}\mathcal{M}$. The derivatives of the chart $\mathbf{h}$ are used to define the standard basis $(\mathbf{e}_1, ..., \mathbf{e}_d) = \frac{\partial \mathbf{h}}{\partial z_1}, ..., \frac{\partial \mathbf{h}}{\partial z_d}$.

The metric tensor $g$ can be used to measure the length of a vector or the angle between two vectors. In local coordinates, the components of the metric tensor are given by

$$G_{ij} = \mathbf{e}_i \cdot \mathbf{e}_j = \frac{\partial \mathbf{h}}{\partial z_i} \cdot \frac{\partial \mathbf{h}}{\partial z_j}, \tag{5}$$

where $\cdot$ is the standard Euclidean scalar product.

The metric tensor $G_{ij}$ describing the manifold $\mathcal{M} \subset \mathbb{R}^D$ learned in manifold learning flows is equivalent to the section of the rectangular Jacobian-transpose-Jacobian $J^T J$ that describes $\mathcal{M}$. As explained in [7] and the appendix, the $J^T J$ can be very efficiently approximated in the context of MLFs. Thus, it can be seen that no additional computational cost is need to calculate the metric tensor.

## 3 Related Work and motivation

Generative models with lower-dimensional latent space have long being established as favorable methods due to their efficiency and expressiveness. For instance, variational autoencoders [19] have been shown to learn data manifolds $\mathcal{M}$ with complex topologies [20, 21]. However, as seen in [22], variational approaches exhibit limitations in learning the distribution $p_{\mathcal{X}}$ on $\mathcal{M}$. Manifold learning flows [6] (Mflow) form a bridge between tractable density and expressiveness. Their support $\in \mathbb{R}^d$ can inherently match the complex topology of $\mathbb{R}^D$, while maintaining a well-defined change of variables transformation. Furthermore, state-of-the-art methods can efficiently calculate the determinant of the Jacobian dot product, as in rectangular normalizing flow [7] (RNF). Denoising normalizing flow [9] (DNF) represents a progression beyond Mflow, where density denoising is used to improve on the density estimation. RNFs can be seen as a parallel approach to DNF but with a more direct methodology, circumventing potential ambiguities stemming from heuristic techniques like density denoising.

Despite the success of manifold learning flows in obtaining a tractable density whilst maintaining expressivity, the learned manifold is not optimized within its latent space representation [23], an outstanding problem for most generative models. For example, due to lack of constrains between latent variables it can be the case that some of them contain duplicate information, i.e., form a degenerate intrinsic basis to describe the manifold, also demonstrated in Section 4.1. Sparse learning [14] was an early attempt to limit this degeneracy and encourage learning of global parameters, introducing solutions such as the relevance vector machine [24, 25] (RVM).The RVM attempts to

limit learning to its minimum necessary configuration, while capturing the most relevant components that can describe major features in the data.

Even stricter approaches are principal [26, 27] and independent component analyses [28, 29], PCA and ICA, respectively. PCA and ICA can obtain a well-defined and optimized representation when convergent; however, PCA and ICA are linear in nature and their non-linear counterparts require kernel definitions. Here, the network-based approach is more general, reducing the need for hand-crafting nonlinear mappings. Furthermore, pure PCA and ICA can be limited in modelling some data manifold topologies where strict global independent components are not necessarily a favorable basis [30]. Furthermore, [15] presents a method for efficient post-learning structuring of the latent space, showcasing the advantages of model compactness. We propose a canonical manifold learning flow where the model is motivated to learn a compact and non-degenerate intrinsic manifold basis.

Furthermore, it has been shown that tabular neural networks can show superior performance when regularized via gradient orthogonalization and specialization [17]. They focus on network regularization, which is not specific to one method but requires certain network structure and costly attribution calculations.

Recently, NF methods with flow manifested in the principal components have been proposed [31, 32]. Cranmer et al. [31] focus on datasets that have distinct PCAs that can be efficiency calculated and, therefore, utilize the computed PCAs for efficient density estimation. Cunningham et al. [32] (PCAflow) relies on a lower-bound estimation of the probability density to bypass the $J^T J$ calculation. This bound is tight when complete principal component flow, as defined by them, is achieved. The fundamental distinction here is that our method does not confine itself to a pure PCAflow scenario, which has the potential to restrict expressivity. In contrast, our method only loosely enforces orthogonality, while encouraging sparsity, i.e. promoting learning of a canonical manifold basis.

## 4 Method

### 4.1 Canonical intrinsic basis

In this section, in order to motivate the proposed development, we first demonstrate how rectangular manifold learning flows assign a latent space representation in comparison to canonical manifold learning flow. We generated synthetic data on a tilted line with noise in the perpendicular and parallel directions, i.e., a fuzzy line. We sample 1000 $x_i$'s such that $x_1 \sim \mathcal{U}(-2.5, 2.5)$ and $x_2 = x_1 + \epsilon$, where $\epsilon \sim \mathcal{U}(-0.5, 0.5)$. In Figure 1(a), a two-dimensional manifold $\mathcal{M}$ is learned with the vanilla-RNF method using the samples by minimizing the loss given in Equation 4. As expected and seen on the left of Figure 1(a), the probability density is learned satisfactorily, i.e., the model fit well the samples and can generate samples with accurate density. However, when samples on the manifold are generated from the two latent dimensions, $z_i$, individually, shown in Figure 1 (ii) and (iii), the sampling results are very similar. The implication being that the intrinsic basis is almost degenerate and information is duplicated amongst the latent variables. In contrast, when the same $\mathcal{M}$ is learned with the proposed CMF, shown in Figure 2, the sampled latent dimensions are close to perpendicular to each other. Each basis is capturing non-degenerate information about $\mathcal{M}$, see Figure 1(b), i.e., the two major axes of the dataset.

### 4.2 Canonical manifold learning

As canonical manifold is not a standard term in literature, let us start by defining a canonical manifold for manifold learning flows.

**Definition 4.1.** *Here, a canonical manifold, $\mathfrak{M}$, is a manifold that has an orthogonal and/or sparse basis $\mathbf{e}_i$ such that $\mathbf{e}_i \cdot \mathbf{e}_j = 0 \; \forall \mathbf{y} \in \mathfrak{M}$ and whenever $i \neq j$.*

The name in Definition 4.1 is inspired from the "canonical basis", i.e., the set of linearly independent generalized eigenvectors of a matrix, as defined in linear algebra. We hypothesize that enforcing learning a canonical manifold as defined in Definition 4.1 during manifold flows will lead to a less degenerate use of the latent space, and consequently a better approximation of the data distribution. As in sparse learning [14], during training necessary dimensions in the latent space will be automatically determined, and those that are necessary will be used in such a way to model a manifold with an

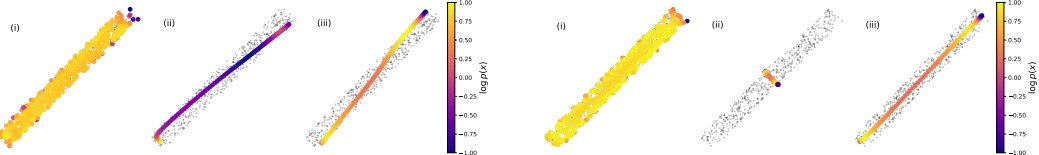

**(a)** Density plot for a fuzzy line learned with RNF, via Equation 4.

**(b)** Density plot for a fuzzy line learned with CMF, via Equation 8.

Figure 1: Comparison of density plots for a fuzzy line learned with RNF (**a**) and CMF (**b**). Sampled 1000 $x_i$'s such that $x_1 \sim \mathcal{U}(-2.5, 2.5)$ and $x_2 = x_1 + \epsilon$ with $\epsilon \sim \mathcal{U}(-0.5, 0.5)$. The black dots represent samples from the real data manifold. (i) Samples from the model using the full latent space where colors correspond to $\log p(x)$. (ii), (iii) Samples from the model using only the first and the second components $z_{1/2} \in \mathcal{Z}$ while setting the other to zero, respectively. Colors correspond to $\log p(x)$ once more. All densities have been normalized to $[-1, 1]$ for visualization purposes. Note that compared to RNF, with CMF different latent variables could capture almost an orthogonal basis for this manifold.

orthogonal local basis to fit the data samples. This, we believe, will enhance the manifold learning process, avoiding overfitting and other learning pathologies associated with correlated attributions to dimensions of the latent space.

In order to obtain a canonical basis we first realize that, as described in Section 2 and the appendix, the metric tensor can be used as concise language to describe the manifold i.e. the transformation of the chart.

$$G_{ij} = \sum_k \frac{\partial g_\phi^{-1,k}(x)}{\partial z^i} \frac{\partial g_\phi^{-1,k}(x)}{\partial z^j} = \sum_k \frac{\partial x^k}{\partial z^i} \frac{\partial x^k}{\partial z^j}. \tag{6}$$

This provides an opportunity to regularize the learned representation directly with a single optimization objective. The dependencies on $G$ are dropped for brevity.

Leveraging the metric tensor representation, we propose to minimize the off-diagonal elements of the metric tensor, enforcing learning of a canonical manifold

$$\|G_{i\neq j}\|_1^1 \triangleq \sum_i \sum_{j\neq i} \|G_{ij}\|_1^1, \tag{7}$$

where $\|\cdot\|_1^1$ is the $\ell_1$ norm. Using $\|G_{i\neq j}\|_1^1$ as a cost to minimize serves both the sparsity and generating orthogonal basis. While the $\ell_2$ norm could be used to enforce a specific embedding [8], it is not in the scope of this work. In order to minimize this cost, the network will have to minimize the dot products $\partial x/\partial z^i \cdot \partial x/\partial z^j$. There are two ways to accomplish this. The first is to reduce the magnitudes of the basis vectors $\partial x/\partial z^i$. Minimizing the magnitudes with the $\ell_1$ norm would lead to sparser basis vectors. The second is to make the basis vectors as orthogonal as possible so the dot product is minimized, which serves towards learning a canonical manifold as well. As detailed in [7] and the appendix, the Jacobian product, the bottleneck for both RNF and our method, can be efficiently approximated. With complexity $\mathcal{O}(id^2)$, it is less than $\mathcal{O}(d^3)$ when $i << d$, where $i$ is the conjugate gradients method iterations. Thus, encouraging a canonical manifold through $G_{i\neq j}$ incurs minimal extra computational cost.

One can imagine an alternative optimization schemes. For instance, the diagonal elements of the metric tensor $G_{kk}$ can be used to access the transformation of individual $z_i \in \mathcal{Z}$. Minimizing the $\ell_1$ norm $\sum_k \|G_{kk}\|_1^1$ will encourage sparse learning, akin to a relevance vector machine [14] or network specialization [17]. Nevertheless, minimizing the diagonal elements will clearly not motivate perpendicular components and notably there is no mixing constraint on these components. Consequently, only the magnitude of the transformation can be affected.

Furthermore, one can imagine minimizing the cosine similarity between the basis vectors to promote orthogonality. However, this would not serve towards automatically determining the necessary dimensions in the latent space. Combining the $\ell_1$ of the diagonal elements and the cosine similarity may also be a solution; however, this would bring an additional weighing factor between the two

losses. $\ell_1$ loss of the off-diagonal elements of the metric tensor brings together these two objectives elegantly.

Note that the injective nature of the flow, the arbitrary topology of the image space and the finite dimensionality of the chart imply that a canonical solution may not exist for the complete learned manifold. However, this is not prohibitive as the local nature of the proposed method should allow even for isolated, sparse and independent component realizations. Even such solutions can be more preferable when representing complex multidimensional image dataset manifolds. Additionally, the canonical solution is not absolutely but rather statistically enforced, i.e. it is not a strict constraint. This is similar to encouraging orthogonality and specialization of gradient attributions for network regularizations [17].

To this end, combining Equation (7) with the established manifold learning flow log likelihood and reconstruction loss from Section 2.1, i.e., Equation (4), we arrive at the following total optimization objective of the canonical manifold learning flow loss:

$$
\phi^* = \operatorname{argmax}_\phi \sum_{i=1}^n \left[ \log p_{\mathcal{Z}} \left( g_\phi^{-1}(x_i) \right) - \log |\det \mathbf{J}_{h_\eta}^T (g_\phi^{-1}(x_i))| \right.
$$
$$
\left. -\frac{1}{2} \log |\det \mathbf{J}_{f_\theta}^T (f_\theta^{-1}(x_i)) \mathbf{J}_{f_\theta}(f_\theta^{-1}(x_i))| - \beta \left\| x_i - g_\phi \left( g_\phi^{-1}(x_i) \right) \right\|_2^2 \right] - \gamma \left\| G_{j \neq k} \right\|_1^1 , \tag{8}
$$

where $\gamma$ is a hyperparameter. In summary, for the transformation parameters $\phi$, the log-likelihood of NFs is maximized, while taking into consideration the rectangular nature of the transformation. Additionally, a reconstruction loss with a regularization parameter $\beta$ is added to ensure that the learned manifold, $\mathcal{M} \subset \mathbb{R}^d$, is aligned to the data manifold, $\mathcal{M} \subset \mathbb{R}^D$. The $\ell_1$ loss of the off-diagonal metric tensor ensures that a canonical manifold $\mathfrak{M}$ is learned.

## 5   Experiments

We compare our method with the rectangular normalizing flow (RNF) and the original Brehmer and Cranmer manifold learning flow (MFlow). No meaningful comparison can be made to linear PCA/ICA as they are not suitable for data on non-linear manifolds. While non-linear PCA/ICA can be used, they require a specific feature extractor or kernel. Our method, a type of manifold learning, directly learns this non-linear transformation from data. We use the same architectures as [7], namely real NVP [7] without batch normalization, for the same reasons explained in the nominal paper [2]. Further experimental details can be found in the appendix. In short, we use the same hyperparameters for all models to ensure fairness. For example, we use a learning rate of $1 \times 10^{-4}$ across the board, and we have only carried out simple initial searches for the novel parameter $\gamma$ from Equation (8). Namely, from trying $\gamma = [0.001, 0.01, 0.1, 1, 10, 50]$, we concluded in using $\gamma = \beta = 1$ for the low-dimensional datasets, $\beta = 5$ and $\gamma = 0.1$ for the tabular datasets and $\beta = 5$ and $\gamma = 0.01$ for the image datasets to ensure stable training. $\gamma = 1$ also produced good results in our experiments with different datasets, but it was not always stable and long training was required. Overall, the method is not overly sensitive to the hyperparameter $\gamma$. Likelihood annealing was implemented for all image training runs. Approximate training times can be found in the appendix.

### 5.1   Simulated data

We consider some simulated datasets. Specifically, we implement uniform distributions over a two-dimensional sphere and a Möbius band embedded in $\mathbb{R}^3$. Then we generate 1000 samples from each of these distributions and use the samples as the simulated data to fit both RNF and CMF. Such datasets have a straightforward and visualizable canonical manifold representation. The samples generated by the trained networks are shown in Figure 2(a) and Figure 2(b) for the RNF and CMF methods, respectively. From the fully sampled density plots Figure 2(a)(i) and Figure 2(b)(i), both methods learn the manifold and the distribution on it. However, the manifold learning of CMF is superior to RNF as the sphere is fully encapsulated, and the density is more uniform. Furthermore, when the three $z_i$ dimensions are sampled individually in Figure 2(b) (ii),(iii) and (iv), the canonical

---
[2]Batch normalization causes issues with vector-Jacobian product computations.

intrinsic basis learned by CMF can be seen. Specifically, two distinct perpendicular components are obtained while the third component approaches zero, i.e. reduced to something small as it is not necessary. In contrast, from Figure 2(a) (ii),(iii) and (iv) it can be seen that in RNF all three components attempt to wrap around the sphere without apparent perpendicularity, and the second component shrinks partially. Even more striking results can be seen for the Möbius band, Figure 2(c) and Figure 2(d). Additionally, the non-trivial topology of the Möbius manifold, the twist and hole, is proving harder to learn for the RNF as compared to CMF. These findings are in direct alignment with the expected canonical manifold learning, where the network is motivated to learn an orthogonal basis and/or implement sparse learning, as explained in Section 4.2. For CMF and RNF methods, the log likelihoods are 1.6553 vs. 1.6517 and KS p-values are 0.17 vs. 0.26. On the sphere, the log likelihoods are 1.97 vs. 1.16. CMF demonstrates better quality than RNF, both quantitatively and qualitatively.

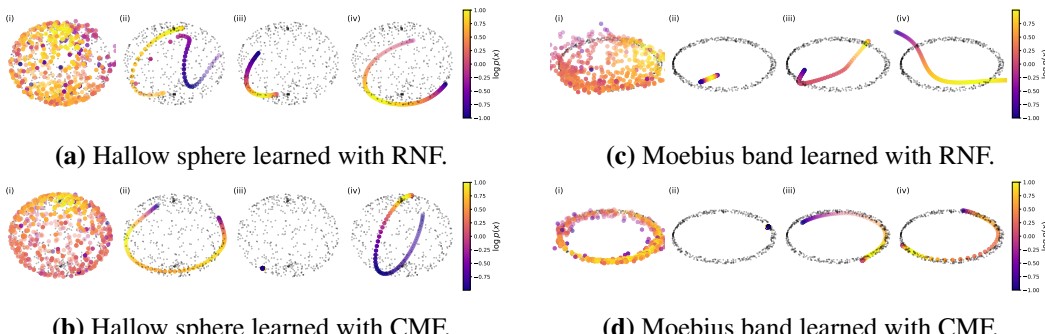

**(a)** Hallow sphere learned with RNF.    **(c)** Moebius band learned with RNF.

**(b)** Hallow sphere learned with CMF.    **(d)** Moebius band learned with CMF.

Figure 2: Density plot for a uniform distribution on a sphere (left) and uniform distribution on a Möbius (right) band learned with RNF (top) and CMF (bottom), the black dots are the data samples, similar to Figure 1. RNF and CMF are learned via Equations 4 and Equation (8) (i) samples from the model using the full latent space where colors correspond to $\log p(x)$. (ii), (iii), (iv) Samples from the model using only the first, the second and third components $z_{1/2/3} \in \mathcal{Z}$ while setting the others to zero, respectively. Colors correspond to $\log p(x)$ once more. All densities have been normalized to $[-1, 1]$ for visualization purposes.

Notably, for these simulated datasets we employ full dimensional latent representations, $D = d$. First, it allows us to visualize what all the latent dimensions are doing in relation to what we expect them to do. In particular, CMF correctly uses only 2 latent dimensions for representing 2D surfaces even though they have 3 latent dimensions. Second, it clearly shows the advantage to RNFs for the case where dimensions of the embedded manifold are known. Such scenario is equivalent to a standard NF. RNF is based on [33], which is an NF that already tries to solve the learning pathologies of complicated manifolds. Therefore, there is no practical limitation in choosing $D = d$, as also seen empirically. Considering the above, the CMF method is showcased to outperform these previous methods, which also aligns with the theoretical intuition.

## 5.2   Image Data

First, Mflow, RNF and CMF are also compared using image datasets MNIST, Fashion-MNIST and Omniglot $\in \mathbb{R}^{784}$. As mentioned, the hyperparameters were set to $\beta = 5$ and $\gamma = 0.01$ for all CMF runs, $\beta = 5$ and $\gamma = 0$ for RNF and Mflow methods, and the network architectures are all equivalent. Furthermore, to emphasize the ability and importance of CMF, we train for different latent dimensions $d$. The test FID scores of the trained models are shown in Table 1. We use $\mathbb{R}^{d=20}$ initially as in [7] for consistency but also run for lower latent dimensions, $\mathbb{R}^{d=10}$, in order to showcase the advantages of CMF relative to the established methods. Namely, a learned canonical intrinsic basis can lead to a more efficient latent space representation and therefore advantageous performance when the latent space is limited. From Table 1, CMF appears to outperform the other methods.

Then, we also experimented using the CIFAR-10 and SVHN datasets with $\mathbb{R}^{3072}$, and CelebA with $\mathbb{R}^{12288}$. Again we set $\beta = 5$ and $\gamma = 0.01$, however, this time with latent dimensions, $\mathbb{R}^{d=30}$, and $\mathbb{R}^{d=40}$. The extra dimensions were necessary for the higher dimensional data. The table Table 1

presents the FID scores. CMF outperforms the other alternatives, consistent with the results obtained on lower dimensional imaging data sets.

In order to visualize the sparsity and orthogonalization encouraged by the CMF, the pair-wise absolute cosine similarity between the basis vectors, i.e., $\{\partial x/\partial z^i\}_{i=1}^d$ (bottom) and the diagonal of the metric tensor (top) are visualized in Figure 3. The CMF clearly induces more sparsity as compared to the RNF, seen from the sparse diagonal elements of the metric tensors, see Figure 3(top), for both the Fashion-MNIST and Omniglot examples. Similarly, the cosine similarity for the CMF is evidently lower for both data sets, see Figure 3(bottom); the MACS stands for the mean absolute cosine similarity. Similar results were observed for the tabular datasets, see appendix.

Additionally, in order to demonstrate the more efficient use of latent space by CMF, we trained both CMF and RNF using latent-dimension $d = 40$ on both the MNIST and the Fashion-MNIST datasets. Then we choose to vary the number of prominent latent dimensions from $\mathcal{Z} \in \mathbb{R}^{40}$ as those with the largest $|G_{kk}|$. We first generate samples from the models using only the prominent dimensions by sampling from the prominent dimensions and setting all others to zero. We compute FID scores for the generated samples. Later, we also reconstruct real samples using only the prominent dimensions and compute reconstruction loss using mean squared error. Here, we take the dimensions, the larger $G_{ii}$ as the ones with the highest weights; this simple interpretation is used for analysis. We repeat the experiment for different number of prominent latent dimensions $(1, 8, 16 \dots, 40)$ non-zero $z_i \in \mathcal{Z}$. In Figure 4(a) and Figure 4(b) the FID score and the mean-squared-error for the different number of prominent dimensions are plotted respectively. From Figure 4, it is evident that the CMF (solid lines) shows better results as compared with RNF (dotted lines) consistently, when using fewer prominent latent dimensions. This suggests that the CMF is able to use the latent space more efficiently. Furthermore, the RNF lines exhibit some oscillation whereas the CMF lines are smoother, indicating that there is also some additional inherent importance ordering in the latent dimensions as expected from sparse learning. Samples generated with different numbers of latent dimensions can be found in the appendix.

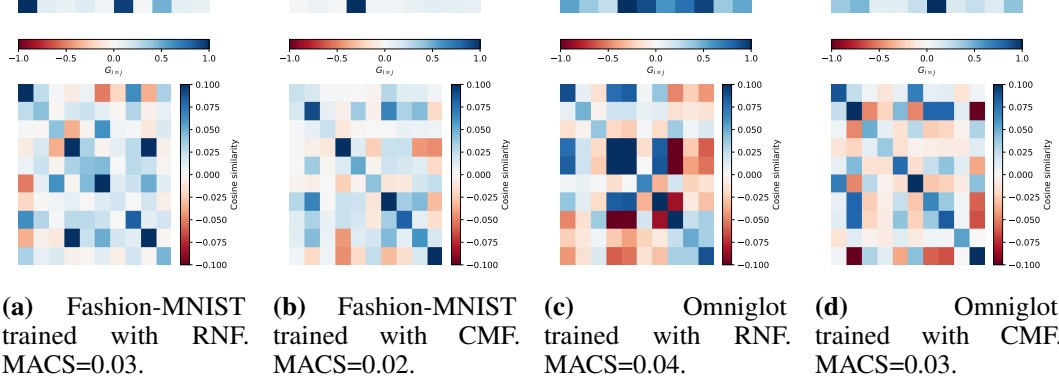

**(a)** Fashion-MNIST trained with RNF. MACS=0.03.

**(b)** Fashion-MNIST trained with CMF. MACS=0.02.

**(c)** Omniglot trained with RNF. MACS=0.04.

**(d)** Omniglot trained with CMF. MACS=0.03.

Figure 3: Sparsity and orthogonalization encouraged by CMF. Average diagonal elements of the metric tensor (top) and average cosine similarity between the basis vectors, $\{\partial x/\partial z^i\}_{i=1}^d$ (bottom), with $\mathbb{R}^{d=10}$. Trained for different datasets with RNF and CMF methods. The sparse metric tensor indicates specialization, and cosine similarity closer to zero indicates orthogonalization. MACS stands for mean absolute cosine similarity. The top plots show the CMF method yielding a sparser transformation, while the bottom ones demonstrate its ability to learn more orthogonal transformations, evidenced further by the reduced MACS values.

## 5.3 Tabular data

The standard normalizing flow benchmarks of tabular datasets from [3] are used to compare the RNF, MFlow and CMF methods. We fit all models with latent dimension $d$=2, 4, 10 and 21 to POWER, GAS, HEMPMASS and MINIBOONE datasets, respectively. The experiments here are similar to those in [7], however, we keep the latent dimension the same for all the methods. Further details can be found in the appendix. We then computed FID-like scores, as was done in compared works [7] to compare the three models. The results are shown in Table 2. CMF appears to have lower FID-like metric scores for POWER, HEMPMASS and MINIBOONE datasets. The GAS dataset

Table 1: Image datasets FID scores,  lower is better.

| Method | MNIST | FMNIST | OMNIGLOT | SVHN | CIFAR10 | CELEBA |
|---|---|---|---|---|---|---|
| | $\mathbb{R}^{d=10}$ | | | $\mathbb{R}^{d=30}$ | | |
| MFlow | 77.2 | 663.6 | 174.9 | 102.3 | 541.2 | N/A |
| RNF | 68.8 | 545.3 | 150.1 | 95.6 | 544.0 | 9064 |
| CMF | **63.4** | **529.8** | **147.2** | **88.5** | **532.6** | **9060** |
| | $\mathbb{R}^{d=20}$ | | | $\mathbb{R}^{d=40}$ | | |
| Mflow | 76.211 | 421.401 | 141.0 | 110.7 | 535.7 | N/A |
| RNF | 49.476 | 302.678 | 139.3 | 94.3 | 481.3 | 9028 |
| CMF | **49.023** | **297.646** | **135.6** | **72.8** | **444.6** | **8883** |

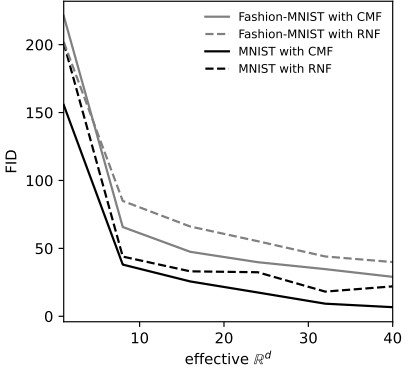

**(a)** Frechet Inception Distance.

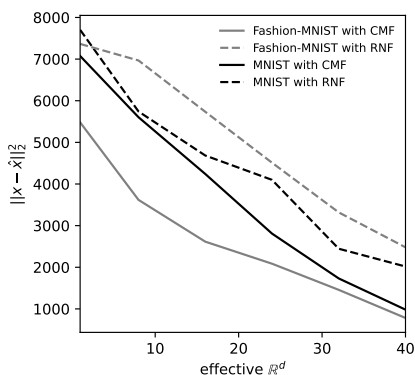

**(b)** Mean-squared-error.

Figure 4: FID test **(a)** score and mean-squared-error **(b)** for fully trained CMF (solid line) and RNF (dotted line) methods on Fashion-MNIST and MNIST with $\mathbb{R}^{d=40}$. The $x-$axis corresponds to sampling from the models using a different number of latent dimensions, while setting others to zero. The chosen latent dimensions are the most prominent ones, defined as those with the largest $|G_{kk}|$ values.

showed mixed results in the second run set, as seen in Table 2. The performance ordering has shifted, suggesting challenges in learning this dataset, possibly leading to random outcomes. However, the other tabular datasets showed consistent performance rankings across run sets. See the appendix for further investigation into the tabular dataset training pathologies. Training on tabular datasets with doubled latent dimensions yielded no significant differences, Table 2.

## 6   Limitations

Our method optimizes the learned manifold $\mathcal{M} \subset \mathbb{R}^d$ of manifold learning flows by constraining the learning to a canonical manifold $\mathfrak{M}$ as in Definition 4.1 at no significant additional computational cost as compared to RNFs [7]. A table of indicative training times can be found in the appendix. Nevertheless, the nature of likelihood-based training, where the full $J^T J$ needs to be calculated, dictates that such a class of methods is computationally expensive for higher dimensions, despite the efficient approximate methods implemented here and in RNFs. Therefore, higher dimensional datasets can be prohibitively computationally expensive to train with this class of methods. Further computational efficiency advancements are expected to be achieved in future works. A notable fact is that the original Mflow method [6] relies on the fundamental approximation such that the full $J^T J$ calculation can be avoided, resulting in inferior log-likelihood calculation and therefore generations, albeit it being noticeably more computationally efficient.

Additionally, CMF can only model a manifold that is a homeomorphism to $R^d$, and when such mapping preserves the topology. Therefore, if, for example, there is a topological mismatch between

Table 2: Tabular data FID-like metric scores, lower is better.

| Method | power | gas (1st run) | gas (2nd run) | hempmass | miniboone |
|---|---|---|---|---|---|
| Mflow | $0.258 \pm 0.045$ | $\mathbf{0.219 \pm 0.016}$ | $0.433 \pm 0.071$ | $0.741 \pm 0.052$ | $1.650 \pm 0.105$ |
| RNF | $0.074 \pm 0.012$ | $0.283 \pm 0.031$ | $0.470 \pm 0.101$ | $0.628 \pm 0.046$ | $1.622 \pm 0.121$ |
| CMF | $\mathbf{0.053 \pm 0.005}$ | $0.305 \pm 0.059$ | $\mathbf{0.373 \pm 0.065}$ | $\mathbf{0.574 \pm 0.044}$ | $\mathbf{1.508 \pm 0.082}$ |
| Double latent dimensions | | | | | |
| RNF | $0.432 \pm 0.022$ | $0.386 \pm 0.044$ | | $0.608 \pm 0.023$ | |
| CMF | $\mathbf{0.205 \pm 0.098}$ | $\mathbf{0.367 \pm 0.007}$ | | $\mathbf{0.519 \pm 0.063}$ | |

the components of the image space $R^D$ described directly by $\mathcal{M} \subset R^d$, it could lead to pathological training. This drawback, however, is shared across all manifold learning flow methods. We also acknowledge the possibility that an orthogonal basis may not always be an optimal representation and difficult to learn: enforcing strict complete orthogonality can restrict the expressivity of the transformation.

Furthermore, other families of state-of-the-art generative modeling methods working in full dimensional representations, for example denoising diffusion models [34], can obtain superior FID scores overall as compared to the class of methods described in this work. However, by default they use a latent space as large as the image space. Notably, for any future use that entails isolating prominent components, it would be beneficial to perform a proper intrinsic dimensionality estimation.

# 7   Summary and Conclusions

In this work, we attempt to improve on the learning objective of manifold learning normalizing flows. Having observed that MLFs do not assign always an optimal latent space representation and motivated from compact latent space structuring [15] and sparse learning [14], while taking advantage of the Riemannian manifold nature of homeomorphic transformations of NFs, we propose an effective yet simple to implement enhancement to MLFs that does not introduce additional computational cost.

Specifically, our method attempts to learn a canonical manifold, as defined in Section 4.1, i.e., a compact and non-degenerate intrinsic manifold basis. Canonical manifold learning is made possible by adding to the optimization objective the minimization of the off-diagonal elements of the metric tensor defining this manifold. We document this learned basis for low-dimensional data in Figure 2 and for image data in Figure 3. Furthermore, we showcase for image data that higher quality in generated samples stems from the preferable, compact latent representation of the data, Table 1.

Distinctly, the off-diagonal manifold metric elements when minimized by an $\ell_1$ loss facilitates sparse learning and/or non-strict local orthogonality. This means that the chart's transformation can be significantly non-linear while retaining some of the benefits of orthogonal representations. This idea can be used in other optimization schemes, which mitigates many of the drawbacks of previous methods in a simple yet theoretically grounded way. Indeed, standard approaches often focus solely on the diagonal elements or strictly enforcing the off-diagonals to be zero, as in [32], limiting expressivity. For instance, in [8], an isometric embedding, essentially a predefined constrained transformation, is presented. Although it allows for direct density estimation and is a form of PCA, it is a strict constraint. Furthermore, as indicated by the synthetic data examples, our present method can be implemented in principle as a full dimensional flow, naturally assigning dimensions accordingly.

The learned manifold representations can be useful as a feature extraction procedure for a downstream task. For example, improved out-of-distribution detection could be a consequence of the feature extraction. Additionally, data that require orthogonal basis vectors, like solutions of a many-body-physics quantum Hamiltonian, can show improved learning performance with the current method.

Manifold learning flows are noteworthy due to their mathematical soundness, particularly in the case of methods such as RNF and CMF, which do not implement any fundamental approximations in the density calculations using lower dimensional latent space.We anticipate that the method will stimulate further research into the nature and importance of the latent space representation of MLFs, enhancing our understanding of generative modelling and paving the way to general unsupervised learning.

## Acknowledgement

This project was supported by grants #2018-222, #2022-274 and #2022-643 of the Strategic Focus Area "Personalized Health and Related Technologies (PHRT)" of the ETH Domain (Swiss Federal Institutes of Technology)

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

# A  Riemannian geometry

The Latin indices run over the spatial dimensions and Einstein summation convection is used for repeated indices.

A $D$ dimensional curved space is represented by a Riemannian manifold M, which is locally described by a smooth diffeomorphism $\mathbf{h}$, called the chart. The set of tangential vectors attached to each point $\mathbf{y}$ on the manifold is called the tangent space $T_{\mathbf{y}}M$. In the fluid model, all the vector quantities are represented as elements of $T_{\mathbf{y}}M$. The derivatives of the chart $\mathbf{h}$ are used to define the standard basis $(\mathbf{e}_1, ..., \mathbf{e}_D) = \frac{\partial \mathbf{h}}{\partial x^1}, ..., \frac{\partial \mathbf{h}}{\partial x^D}$.

The metric tensor $g$ can be used to measure the length of a vector or the angle between two vectors. In local coordinates, the components of the metric tensor are given by

$$g_{ij}(x) = \mathbf{e}_i(x) \cdot \mathbf{e}_j(x) = \frac{\partial \mathbf{h}}{\partial x^i} \cdot \frac{\partial \mathbf{h}}{\partial x^j}, \tag{9}$$

where $\cdot$ is the standard Euclidean scalar product.

For a given metric tensor, the vector $v = v^i \mathbf{e}_i \in T_{\mathbf{y}}M$ has a norm $||v||_g = \sqrt{v^i g_{ij} v^j}$ and a corresponding dual vector $v^* = v^i \mathbf{e}_i \in T_{\mathbf{y}}^*M$ in the cotangent space, which is spanned by the differential 1-forms $dx^i = g(\mathbf{e}_i, \cdot)$. The coefficients $v_i$ of the dual vector are typically denoted by a lower index and are related to the upper-index coefficients $v^i$ by contraction with the metric tensor $v_i = g_{ij} v^j$ or equivalently, $v^i = g^{ij} v_j$, where $g^{ij}$ denotes the inverse of the metric tensor. The upper-index coefficients $v^i$ of a vector $v$ are typically called *contravariant components*, whereas the lower-index coefficients $v_i$ of the dual vectors $v^*$ are known as the *covariant components*.

A necessary feature for the description of objects moving on the manifold is parallel transport of vectors along the manifold. The tangent space is equipped with a covariant derivative $\nabla$ (Levi-Civita connection), which connects the tangent spaces at different points on the manifold and thus allows for the transportation of a tangent vector from one tangent space to the other along a given curve $\gamma(t)$. The covariant derivative can be viewed as the orthogonal projection of the Euclidean derivative $\partial$ onto the tangent space, such that the tangency of the vectors is preserved during the transport. In local coordinates, the covariant derivative is fully characterized by its connection coefficients $\Gamma^i_{jk}$ (Christoffel symbols), which are defined by the action of the covariant derivative on the basis vector, $\nabla_j \mathbf{e}_k = \Gamma^i_{jk}$. In the standard basis, $\mathbf{e}_i = \frac{\partial \mathbf{h}}{\partial x^i}$, the Christoffel symbols are related to the metric by

$$\Gamma^i_{jk} = \frac{1}{2} g^{ij} (\partial_j g_{kl} + \partial_k g_{jl} - \partial_l g_{jk}). \tag{10}$$

Acting on a general vector $v = v^i \mathbf{e}_i$, the covariant derivative becomes:

$$\nabla_k v = (\partial_k v^i + \Gamma^i_{kj} v^j) \mathbf{e}_i, \tag{11}$$

where the product rule has been applied, using that the covariant derivative acts as a normal derivative on the scalar functions $v^i$. Extended to tensors of higher rank, for example the second order tensors $T = T^{ij}$, it becomes

$$\nabla_k T = (\partial_k T^{ij} + \Gamma^i_{kl} T^{lj} + \Gamma^j_{kl} T^{il}) \mathbf{e}_i \otimes \mathbf{e}_j \tag{12}$$

; in this work, the basis vectors $\mathbf{e}_i$ are generally dropped. Compatibility of the covariant derivative with the metric tensor implies that $\nabla_k g^{ij} = \nabla_k g_{ij} = 0$. This property allows us to commute the covariant derivative with the metric tensor for the raising or lowering of tensor indices in derivative expressions.

The motion of the particle can be described by the curve $\gamma(t)$, which parameterizes the position of the particle at time $t$. The geodesic equation, $\nabla_{\dot{\gamma}} \dot{\gamma} = 0$, in local coordinates $\gamma(t) = \gamma^i(t) \mathbf{e}_i$ is defined by

$$\ddot{\gamma}^i + \Gamma^i_{jk} \dot{\gamma}^j \dot{\gamma}^k = 0. \tag{13}$$

The geodesic equation can be interpreted as the generalization of Newtons law of inertia to curved space. The solutions of Eq. (13) represent lines of constant kinetic energy on the manifold, i.e. the geodesics. The Riemann curvature tensor $R$ can be used to measure curvature, or more precisely, it measures curvature-induced change of a tangent vector $v$ when transported along a closed loop.

$$R(\mathbf{e}_i, \mathbf{e}_j)v = \nabla_i \nabla_j v - \nabla_j \nabla_i v. \tag{14}$$

In a local coordinate basis $\mathbf{e}_i$, the coefficients of the Riemann curvature tensor are given by

$$R^l_{ijk} = g(R(\mathbf{e}_i, \mathbf{e}_j)\mathbf{e}_k, \mathbf{e}_l) \tag{15}$$

$$= \partial_j \Gamma^l_{ik} - \partial_k \Gamma^l_{ij} + \Gamma^l_{jm}\Gamma^m_{ik} - \Gamma^l_{km}\Gamma^m_{ij}. \tag{16}$$

Contraction of $R^i_{jkl}$ to a rank 2 and 1 tensor yields the Ricci-tensor $R_{ij} = R^k_{ikj}$ and the Ricci-scalar $R = g^{ij}R_{ij}$ respectively, which can also be used to quantify curvature.

The gradient is defined as $\nabla^i f = g^{ij}\partial_j f$, the divergence as $\nabla_i v^i = \frac{1}{\sqrt{g}}\partial_i(\sqrt{g}v^i)$, and the integration over curved volume as $V = \int_V Q dV$, where $dV = \sqrt{g}dx^1...dx^D =: \sqrt{g}d^D x$ denotes the volume element. $\sqrt{g}$ denotes the square root of the determinant of the metric tensor.

## B  Extension to Related Work and further discussions

Normalizing flows have direct access to the latent distribution due to homeomorphism defining the transformation [4]. Nevertheless, the very nature of the homeomorphism can result in pathological behaviour when manifolds with complex topology are learned. For instance, a two-dimensional data space $p_\mathcal{X}$ can have different numbers and types of connected components, 'holes' and 'knots', and images have complicated shapes and topology due to the sheer number of dimensions. Therefore, as usually $p_\mathcal{Z}$ is modelled by a simple distribution, like a Gaussian, it cannot trivially match the topology of $p_\mathcal{X}$. [33] demonstrates a stable solution to this topological pathology of NFs, albeit with $\mathbb{R}^D$ support.

Effectively learning manifolds with complicated topologies remains a challenge, even with the full-dimensional latent space. However, the RNF is based on CIF [33] which is an NF that already tries to solve the learning pathologies of complicated manifolds: "CIFs are not subject to the same topological limitations as normalizing flows" [33]. Furthermore, in that regard, we estimate there is no practical limitation in setting, as also seen empirically. In principle, CMF can be a solution for higher-dimensional data. However, in practice it is very computationally expensive to train a full latent dimension (e.g. $J^T J$ calculation) and the solution will take long to converge. Additionally, empirical knowledge suggests that a lower-dimensional representation can enhance expressivity, as discussed in the context of Mflows. Consequently, in practice using the current method, one can start from a predefined lower dimension and let the network optimize at will.

PCAflow [32] relies on a lower-bound estimation of the probability density to bypass the $J^t J$ calculation. This bound is tight when complete principal component flow, as defined by them, is achieved. The fundamental distinction here is that CMF does not confine itself to a pure PCAflow scenario, which has the potential to restrict expressivity. Since our method only loosely enforces orthogonality, or mutual information $\rightarrow 0$, we anticipate the contour lines to lie somewhere between NF and PCAflow. Furthermore, the introduction of sparsity adds an extra layer of complexity, making direct comparison challenging. As an outlook, we acknowledge that DNF [9] and PCAflow, can undergo quantitative comparison with our method. However, we estimate that RNF is precisely in alignment with CMF, providing a more representative comparison.

Generally, the meaning of "canonical manifold" can vary depending on the context in which it is used, therefore we use Definition 4.1 to precisely define the term "canonical manifold". Definition 4.1 aligns closely with the assumption of PCAflow. However, it is vital to note that our method does not enforce this strictly. To put it in perspective, our approach seeks a 'partly canonical manifold,' if you will. Additionally, our method encompasses sparsity—meaning the diagonal elements can also approach zero.

In principle, full-dimensional flow $d = D$ can be also solution for higher-dimensional datasets, as the CMF loss will ensure that the number of prominent latent dimensions should correspond to the true intrinsic dimensionality of the data. However, in practice it is very computationally expensive to train a full latent dimension (e.g. $J^t J$ calculation) and the solution will take long to converge. Additionally, empirical knowledge suggests that a lower-dimensional representation can enhance expressivity, as discussed in the context of Mflows. Consequently, in practice, one can start from a predefined lower-dim and let the network optimize at will. Nevertheless, the efficient approximate methods indicate that CMF could be implemented for a full-dimensional flow. Further exploration of this potential could be undertaken in future studies.

## C  FID score

For a dataset $\{x_1, \ldots x_n\} \subset \mathbb{R}^D$ and sampled generated data $\{\tilde{x}_1, \ldots \tilde{x}_n\} \subset \mathbb{R}^D$ for a statistic $F$ the mean and variance are:

$$\mu = \frac{1}{n}\sum_1^n F(x_i), \ \Sigma = \frac{1}{n-1}\sum_1^n (F(x_i) - \mu)(F(x_i) - \mu)^T, \tag{17}$$

$$\tilde{\mu} = \frac{1}{n}\sum_1^n F(\tilde{x}_i), \ \tilde{\Sigma} = \frac{1}{n-1}\sum_1^n (F(\tilde{x}_i) - \tilde{\mu})(F(\tilde{x}_i) - \tilde{\mu})^T. \tag{18}$$

The FID score takes $F$ as the last hidden layer of a pre-trained network and evaluates generated sample quality by comparing generated moments against data moments. This is achieved by calculating the squared Wasserstein-2 distance between Gaussian and corresponding moments by:

$$\|\mu - \tilde{\mu}\|_2^2 + tr(\Sigma + \tilde{\Sigma} - 2(\Sigma\tilde{\Sigma})^{1/2}. \tag{19}$$

This would reach zero if they match. For tabular data, $F$ is taken to be the identity $F(x) = x$, as in [7].

## D  Optimizing the objective and calculating the metric tensor

The third term of canonical manifold learning optimization objective, (Equation (8)):

$$\phi^* = \mathrm{argmax}_\phi \sum_{i=1}^n \Bigg[ \log p_{\mathcal{Z}}\left(g_\phi^{-1}(x_i)\right) - \log|\mathrm{det}\mathbf{J}_{h_\eta}^T(g_\phi^{-1}(x_i))|$$

$$-\frac{1}{2}\log|\mathrm{det}\mathbf{J}_{f_\theta}^T(f_\theta^{-1}(x_i))\mathbf{J}_{f_\theta}(f_\theta^{-1}(x_i))| - \beta\left\|x_i - g_\phi\left(g_\phi^{-1}(x_i)\right)\right\|_2^2 \Bigg] - \gamma\|G_{j\neq k}\|_1^1,$$

is not so trivial to backpropagate through. Here we use the method suggested by [7]. First, $J_\theta^T J_\theta$ is calculated efficiently with the Forward-Backward automatic differentiation trick as explained in Section 4.3 of [7]. For the simulated low-dimensional datasets the exact method is used, and for images the approximate stochastic gradient method with $K = 1$ is used. Second, to back-propagate for the approximate method, a stochastic unbias estimator is used. Namely, from matrix calculus [35] it is known that:

$$\frac{\partial}{\partial\theta_j}\mathrm{logdet}J_\theta^T J_\theta = tr[(J_\theta^T J_\theta)^{-1}\frac{\partial}{\partial\theta_j}J_\theta^T J_\theta], \tag{20}$$

where $tr$ is the trace and $\theta_j$ is the $j$-th component. Then the Hutchinson trace estimator is used [36]:

$$\frac{\partial}{\partial\theta_j}\mathrm{logdet}J_\theta^T J_\theta \approx \frac{1}{K}\sum_{k=1}^K \epsilon_k^T(J_\theta^T J_\theta)^{-1}\frac{\partial}{\partial\theta_j}J_\theta^T J_\theta\epsilon_k, \tag{21}$$

where $\varepsilon_1, \ldots \varepsilon_k$ are sampled from a Gaussian distribution. Access to $J_\theta^T J_\theta$ as explained above allows for the direct computation of Equation (21). Nevertheless, the first term $\epsilon_k^T(J_\theta^T J_\theta)^{-1} = [(J_\theta^T J_\theta)^{-1\varepsilon}]^T$ is calculated by an iterative method solver of the matrix inverse problem, the conjugate gradient method $CG$ [37].

$$\frac{\partial}{\partial\theta_j}\mathrm{logdet}J_\theta^T J_\theta \approx \frac{1}{K}\sum_{k=1}^K CG((J_\theta^T J_\theta; \epsilon_k)^T\frac{\partial}{\partial\theta_j}J_\theta^T J_\theta\epsilon_k. \tag{22}$$

Lastly, the metric tensor is obtained from $J_\theta^T J_\theta$ as stated in Section 2.2. Back-propagation of $\|G_{j\neq k}\|_1^1$ is straight forward by taking the derivative w.r.t. $\theta$ by an (E.g. Adams) optimizer, similar to the reconstruction loss in Equation (8).

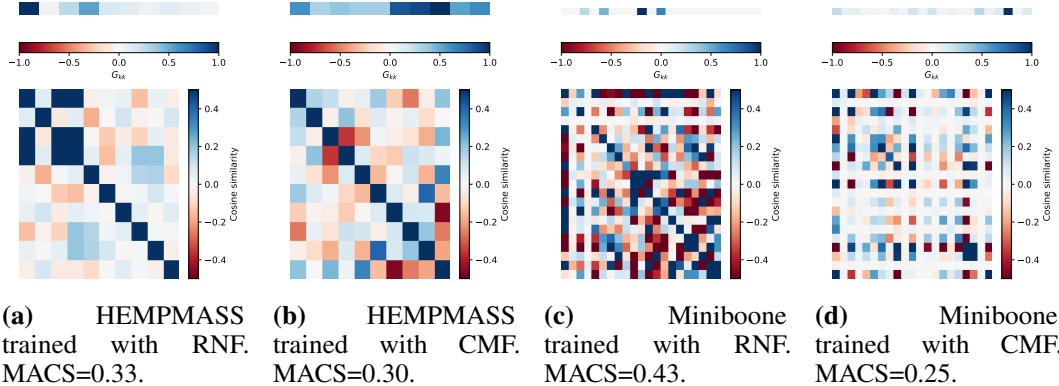

**(a)** HEMPMASS trained with RNF. MACS=0.33.

**(b)** HEMPMASS trained with CMF. MACS=0.30.

**(c)** Miniboone trained with RNF. MACS=0.43.

**(d)** Miniboone trained with CMF. MACS=0.25.

Figure 5: Sparsity and orthogonalization encouraged by CMF. Average diagonal elements of the metric tensor (top) and average cosine similarity between the basis vectors, $\{\partial x/\partial z^i\}_{i=1}^{d}$ (bottom), with $\mathbb{R}^{d=10}$. Trained for different tabular datasets with RNF and CMF methods. The sparse metric tensor indicates specialization, and cosine similarity closer to zero indicates orthogonalization. MACS stands for mean absolute cosine similarity.

# E  Tabular datasets

In Figure 5, we visualize the metric tensor for the tabular datasets trained with sufficiently high latent dimensions and present the mean absolute cosine similarity (MACS) in the supplementary PDF. For the specific case of GAS with d=2, the plot might not offer significant insights. We observe that the mean absolute cosine similarity is lower for CMF, although, considering the error, there remains some overlap, which is within expectations.

In order to investigate the cause for the performance gap in GAS, we visualize the validation FID-like scores during training, Figure 6. Upon comparing the results of HEMPMASS and GAS, it is apparent that learning converges until a certain point, after which instability may arise for all three methods. Despite this instability, it is important to note that our implementation selects the best validation step checkpoint for testing, mitigating the impact of training instability. However, it is plausible that these instabilities contribute to the variations between different runs. Furthermore, the training curves follow similar behaviors, converging up to a certain point and then displaying a degree of instability. This trend holds true for all three methods when trained on the GAS dataset.

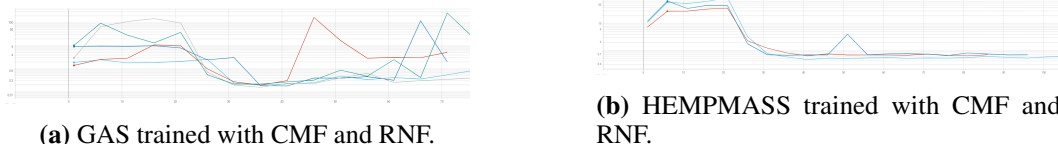

**(a)** GAS trained with CMF and RNF.

**(b)** HEMPMASS trained with CMF and RNF.

Figure 6: FID-like validation score training curves. Different colors correspond to different repetitions with both methods, CMF and RNF.

# F  Image generations

## F.1  Samples from prominent dimensions

In order to demonstrate the more efficient use of latent space by CMF, we trained both CMF and RNF using latent-dimension $d = 40$ on both MNIST and Fashion-MNIST datasets. We choose a varying number of prominent latent dimensions from $\mathcal{Z} \in \mathbb{R}^{40}$ as those with the largest $|G_{kk}|$, this is mathematical intuition that the largest eigenvalues (if an eigen-decomposition is possible) correspond to the dimensions with the highest weights.

We generate samples from the models using only the prominent dimensions, by sampling from the prominent dimensions and setting all others to zero, subsequently FID scores are calculated for the

generated samples. Additionally, we reconstruct real samples using only the prominent dimensions and compute reconstruction loss using mean squared error, as shown in the main text. We repeat the experiment for different number of prominent latent dimensions $(1, 8, 16 \ldots, 40)$, or similarly, non-zero $z_i \in \mathcal{Z}$. Here, in the appendix, we show samples when the number of prominent dimensions sampled from increases.

We would like to note that potentially for any future use that entails isolating prominent components, a proper ID estimation should be carried out.

For example, in Figure 7, different samples are generated along the horizontal. This is repeated for increasing number of prominent dimensions used for sampling, top to bottom along the vertical $(4, 8, 12, 16 \ldots, 40)$. In Figure 7, FashionMNIST models were trained with $\mathbb{R}^{d=40}$ and for the CMF and RNF methods. Similarly to the FID plots in the main text, the CMF shows a bit less variation when going along the columns from top to bottom; this is an indication that fewer prominent dimensions are needed for a faithful generation.

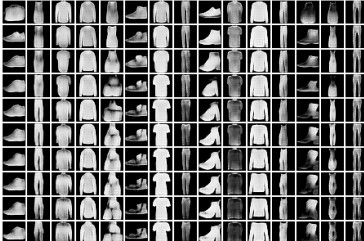 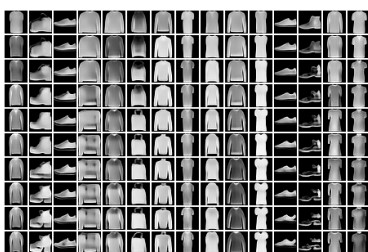

(a)                                                        (b)

Figure 7: Generated samples for increasing prominent dimensions used for sampling from top to bottom along the columns. The model is trained on Fashion-MNIST with $\mathbb{R}^{d=40}$, (a) RNF, (b) CMF. Rows represent different samples. The CMF captures the generations almost fully, even with the first prominent dimensions.

To visualize the sparsity of learned manifold, we generate samples from individual latent space dimensions for fully trained models on FashionMNIST with the RNF and CMF. In Figure 8, the trained RNF and CMF models exhibit FIDs of $\sim 302$ and $297$ respectively, see main text. Across the rows, different samples can be found as sampled from each of 10 prominent component subgroups, i.e. $(\{1, 2\}, \{3, 4\}, \{5, 6\} \ldots, \{19, 20\})$ out of the 20 latent dimensions. Here, only these subgroups are used for sampling at once and the other 18 dimensions are set to zero. The dimensions sampled from at the bottom are the most significant ones; the rest follow in order towards the top. It can be clearly seen that the CMF method manages to capture most of the information of the manifold in the first two dimension, as the generations in the bottom row are good and the rest of the columns seem to have less relevance, indicating strong sparse learning. For the RNF, again there is an importance distinction, as expected, but even dimensions of less prominence seem to be generating variation in the samples. Combining these observations with the higher FID of the CMF, efficient sparse learning is evident.

In Figure 9, we plot a sampled manifold between -3 and 3 for the two most prominent latent dimensions from $\mathcal{Z} \in \mathbb{R}^{20}$ as those have the largest $|G_{kk}|$. This is done for an RNF and CMF trained MNIST model on the left and right of Figure 9. The CMF method appears to split the latent space of $z_1$ without much variation in the $z_2$ direction, indicating perpendicularity.

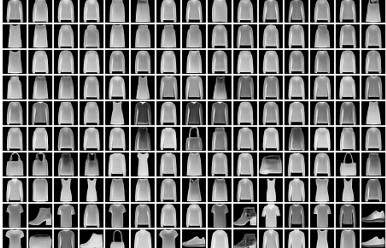 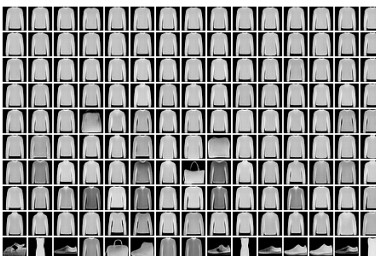

(a)                                                                    (b)

Figure 8: Generated samples for 10 different hierarchical subgroups of prominent dimensions $(\{1,2\}, \{3,4\}, \{5,6\} \ldots, \{19,20\})$ along the columns from bottom to top, while setting all other dimensions to zero. Models were trained on Fashion-MNIST with $\mathbb{R}^{d=20}$ with (a) RNF, (b) CMF. Rows represent different samples. Sparse learning is evident for the CMF method.

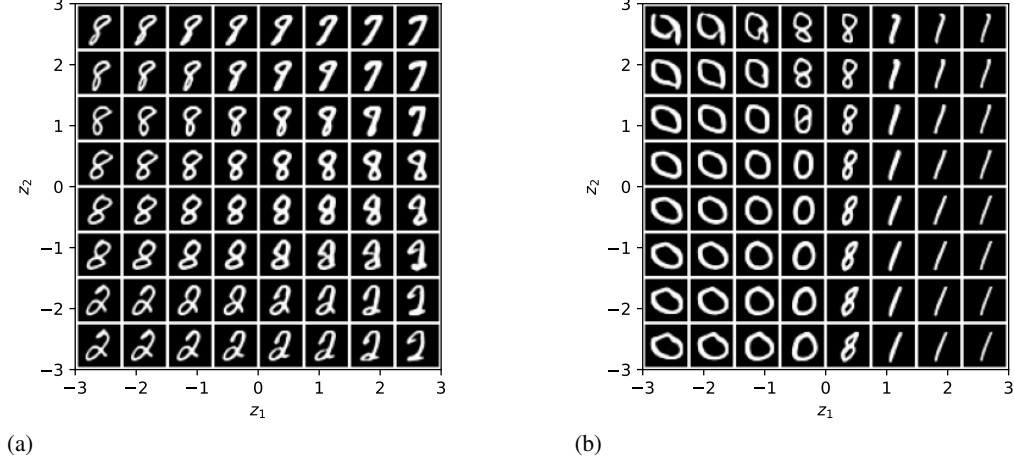

(a)                                                                    (b)

Figure 9: Sampled manifold for the two most prominent dimensions as calculated from $G_{kk}$, (a) RNF, (b) CMF.

## F.2 Out of distribution detection

We evaluate the performance of CMF for out-of-distribution detection (OoD). Table 3 quantifies the results, where accuracy here is the log-likelihood. The results are similar for the previous methods as expected.

## G Experimental details

The training was performed on a GPU cluster with various GPU nodes including Nvidia GTX 1080, Nvidia GTX 1080 Ti, Nvidia Tesla V100, Nvidia RTX 2080, Nvidia Titan RTX, Nvidia Quadro RTX 6000, Nvidia RTX 3090. Multiple GPUs were used in parallel across the batch.

Table 3: Decision stump for OoD accuracy (higher is better)

| METHOD | MNIST → FMNIST | FMNIST → MNIST |
|--------|----------------|----------------|
| M-FLOW | 94 % | 84 % |
| RNF | 96 % | 78 % |
| CMF | 97 % | 77 % |

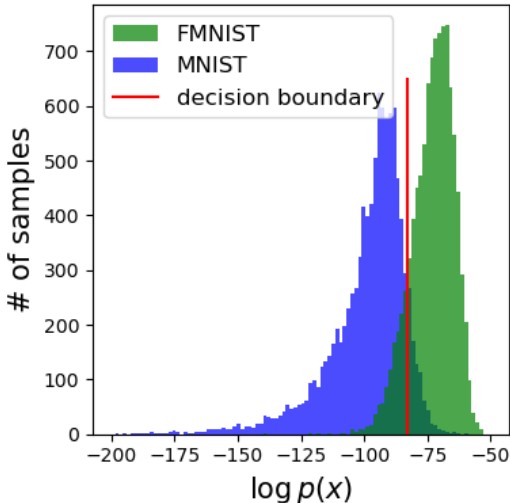

Figure 10: OoD dection with CMF, trained on Fashion-MNIST.

There exists no substantial computational cost distinction between the RNF and CMF methods, as both encounter the calculation bottleneck of the Jacobian-transpose-Jacobian. Any other option was set to the default as found in [7]. The approximate training times can be found in Table 4. Currently, manifold learning flows are computationally demanding; nevertheless, their potential is noteworthy due to their mathematical soundness, particularly in the case of methods such as RNF and CMF, which do not implement any fundamental approximations in the density calculations using lower dimensional latent space.

Table 4: Approximate training times for $\mathbb{R}^{d=40|20}$. D and H indicate days and hours respectively. MNIST and FMNIST required around 120 epochs, SVHN, around 350, CelebA around 220, CIFAR10 around 500 epochs of training.

| METHOD | SVHN | CIFAR10 | CELEBA | MNIST | FMNIST |
|--------|------|---------|--------|-------|--------|
| M-FLOW | N/A | N/A | N/A | 1D | 20H |
| RNF | 12D | 20D | 28D | 3D | 2D 18H |
| CMF | 14D | 21D | 27D | 3D 4H | 2D 17H |

## G.1 Images

To train for images, 4 or 5 nodes of less than 10GB each were used in parallel across the batch size. The hyperparameters were kept constant for the three different methods compared, unless a distinction was made for the manifold learning flow as explained in [7].

| Parameter | Setting |
| --- | --- |
| Learning rate | $1 \times 10^{-4}$ |
| Optimizer | Adams |
| Likelihood annealing | between epochs 25 and 50 |
| Early stopping | MNIST, FashionMNIST, Omniglot: min 121 epochs |
| | SVHN, CelebA, CIFAR10: min 201 epochs |
| Batch size | 120 |
| Latent dimensions | $\{5, 10, 20, 30, 40\}$ |
| Hyperparameters | Equation (8): $\beta = 5, \gamma = 0.1, 0.01$ |
| Calculation method | Jacobian-transpose-Jacobian: Hutchinson, Gaussian, K=1 |
| CG tolerance | 0.001 |
| D-dim flow coupler | 8x64 |
| d-dim flow layers | 10 |

## G.2   Simulated data

The parameter choice is also kept consistent for training for simulated data. Training was performed on a single GPU or CPU and training times were observed to be 1-4 hours depending on epochs and if early stopping was used. Again, similar training times between RNF and CMF were observed.

| Parameter | Setting |
| --- | --- |
| Learning rate | $1 \times 10^{-4}$ |
| Optimizer | Adams |
| Likelihood annealing | False |
| Epochs | 5000 (without early stopping) |
| Batch size | 1000 |
| Latent dimensions | $\{2, 3\}$ |
| Hyperparameters | Equation (8): $\beta = 1, \gamma = 1$ |
| Calculation method | Jacobian-transpose-Jacobian: 'exact' |
| D-dim flow coupler | RealNVP |
| d-dim flow layers | 4 |

## G.3   Tabular data

We train for the tabular datasets as found in Papamakrios et al. [3]. Parameters and settings were chosen as follows, as per RNF.

| Parameter | Setting |
| --- | --- |
| Learning rate | $1 \times 10^{-4}$ |
| Optimizer | Adams |
| Likelihood annealing | between epochs 25 and 50 |
| Early stopping | Yes |
| Batch size | POWER: 5000, GAS: 2500, HEMPMASS: 750 ,MINIBOONE: 400 |
| Hyperparameters | Equation (8): $\beta = 5, \gamma = 0.01$ |
| Calculation method | Jacobian-transpose-Jacobian: Hutchinson, Gaussian, K=1 |
| CG tolerance | 0.001 |
| D-dim flow coupler | 4x128 with 10 layers |
| d-dim flow | RealNVP 2x32 with 5 layers |
| Latent dimension | d=D/2 |

For each run set, we performed 5 repetitions, of which we report an average FID score and a standard error in Table 2. We train models with latent dimension d=2, 4, 10 and 21 to POWER, GAS, HEMPMASS and MINIBOONE datasets, respectively.

