# OpenReview forum: "Canonical normalizing flows for manifold learning"
_NeurIPS.cc/2023/Conference — NeurIPS 2023 poster_

### Official Review · Reviewer_iKjw · 2023-06-14

**Soundness:** 2 fair
**Presentation:** 2 fair
**Contribution:** 3 good
**Rating:** 6
**Confidence:** 3

**Summary:**

The authors address the interesting question of how to disentangle the relevant manifold directions properly in the latent space of manifold learning normalizing flows (MLF). After a comprehensive theoretical background, the main motivation is presented in form of a toy example of a simple noisy line embedded in $\mathbb{R}^{2}$. When varying the latent dimension $z_{i}$ of a RNF and setting the remaining latent variables to $0$, the corresponding contour lines are partially aligned. Doing the same with the proposed method, CMF, leads to orthogonal contour lines and thus to a meaningful latent representation where one latent variable encodes the line and the other the (noisy) off-manifold direction. This intuition leads straightforwardly to enforcing the orthogonality of the flows Jacobian matrix. To do so, the authors propose a new cost function for learning manifold structures using an NF by adding a penalty term which ensures that the off-diagonal entries of the Gram matrix (Jacobian-transpose-Jacobian) are $0$ at the one hand, and are sparse at the other due to the l1-norm. The effecitiveness of the method is tested in experiments on toy-data, tabular data and 32x32x3 image data.


**Strengths:**

The paper is overall well written. The theoretical background is exhaustive and well-explained. The example in 4.1 serves as a good illustrative motivation. The experiments are comprehensive.

**Weaknesses:**

The contribution is somewhat marginal. Though citing [1], they don't mention that the corresponding authors propose a very similar penalty term. To my understanding, the only differences of the penalty term in [1] are
a) that in [1] the diagonal entries are penalized to be $1$ whereas in this work the diagonal entries are unconstrained.
b) [1] used the L2-norm whereas in this paper the L1-norm is used.

Given the great similarity, the penalty term suggested by [1] should be included in the comparison. What do we gain by not constraining the flow to be an isometry? In addition, the authors use the M-flow for benchmarking although a better method for overcoming the limitations of M-flows (namely that the log-determinant term is not considered while training) was already proposed in [2]. At least citing and ideally comparing with [2] should be part of the paper. The same is true for comparing with [3]. It would be nice to see how the contour lines differ from the method introduced in [3].

Furthermore, I am not sure to understand the reasoning behind Definition 4.1. To my understanding, every differentiable manifold is a canonical manifold as one is free to choose the basis of the tangent space to be orthogonal. Thus, such a basis always exists which is the only requirement to be a canonical manifold. Definition 4.1. is simply an existence statement and not useful as such. I think what the authors wanted to define is a manifold equipped with a chart s.t. the chart induces an orthogonal basis of the tangent space. However, this is very much the definition of the principal component flow [3]. Please enlighten me in case I totally misunderstood something.

I am also confused about the comparison with rectangular NF (RNF) in Figure 2. By definition for RNF we have that $d<D$. However, in Figure 2 the author uses $d=D$. This is a contradiction. If $d=D$, a standard NF can be used rather than an RNF. In addition, one claimed strength of the proposed CMF is the sparisity induced by the l1 norm. Thus, the relevant dimensions should be found automatically which elevates, in theory, the necessity to estimate $d$ a priori. Then, why not always set $d=D$ and then apply the proposed penalty term? The number of prominent latent dimensions should correspond to the true intrinsic dimensionality of the data.

Finally, the notation for the manifold, data manifold, and learned manifold is confusing. In line 74 or line 90, $\mathcal{M}$ is a lower dimensional manifold. In line 83, $\mathcal{M}_{\theta}$ is introduced without explaination. In line 212, $\mathcal{M}$ is referred to as the learned manifold and in the same line, the data manifold is introduced without further explanation. In addition, a new notation for a canonical manifold is suggested. I find the various versions for a manifold confusing and don't see the added benefit of it (especially given that Definition 4.1. is not useful in my opinion).


[1] Eike Cramer, Felix Rauh, Alexander Mitsos, Raúl Tempone, and Manuel Dahmen. Nonlinear 350 isometric manifold learning for injective normalizing flows. arXiv preprint arXiv:2203.03934, 351 2022.

[2] C. Horvat and J.-P. Pfister. Denoising normalizing flow. In Advances in Neural Information Processing Systems, volume 34, 2021.

[3] Edmond Cunningham, Adam D Cobb, and Susmit Jha. Principal component flows. In Interna410 tional Conference on Machine Learning, pages 4492–4519. PMLR, 2022.


**Questions:**

1. Does Definition 4.1 really makes sense (see above)
2. Is the comparison with RNF in Figure 2 fair (see above)? By definition of RNF, it must be $d<D$. Please elaborate.
3. Why not always set $d=D$ and then apply the proposed penalty term (see above)?
4. Figure 1. Why is the density varying greatly when only varying one dimension? Shouldn't be uniform if the density is properly learned?
5. What is the quality in terms of density estimation? For instance, what is the KS statistics compared to the true density, or, at least, does it sum to 1?
6. Why not benchmark with PCA flow? It would be interesting to see if this method can overcome some of its limitations (see above).
7. Moebius band with RNF seems to be poorly learned. Why?
8. Why prominent latent dimensions are the ones with the greatest $G_{ii}$? Is this a mathematical statement/proposition or intuition?
9. line 305: in which case would an orthogonal basis not be an optimal representation?
10. What happens if we use the L2 rather than L1 norm?
11. How are prominent latent dimensions defined? Do they correspond to the true intrinsic dimensionality? Is there a clear cut-off in the magnitudes of $|G_{kk}|$ in e.g. MNIST? If not, a somewhat arbitrary threshold needs to be defined, isn't it?

*Minor*:
+ line 70-72: the statement is only true for large $n$
+ line 83: What is $\mathcal{M}_{\theta}$?
+ line 83: "to encorage $x\in \mathcal{M}_{\theta}$" do you mean $g_{\phi}(z)$ rather than $x$?
+ line 114: typo: "due to lack..."
+ line 145: noise is not perpendicular to line
+ line 217: mention reason (maybe in a footnote)
+ line 225: $\gamma$ not crucial but had an impact on training time? What is crucial for you?
+ line 244: bad wording "the expected the canonical manifold learning"
+ line 297: Inconsistent acronym: M-flow vs. Mflow
+ line 299: inferior lok-likelihood calculations of Mlow...where shown?
+ line 309/310: wrong statement: there are diffusion models with lower dimensional latent space. However, by default indeed the latent space has the same dimensionality

**Limitations:**

Major limitations are addressed in Section 6. However, some questions are unresolved such as: how to find/define the prominent latent dimensions without an intrinsic dimensionality estimator?

---

> ### Author Rebuttal · Authors · 2023-08-09
>
> W1: We appreciate the reviewer for their insightful comments. We would like to emphasize that calculating L1 for the off-diagonal elements is not a trivial conceptual step while it may seem like a simple implementation detail, and this distinction mitigates many of the drawbacks of previous methods in a simple yet theoretically grounded way. For instance, in [1], an isometric embedding —essentially a predefined constrained transformation— is presented. Although it allows for direct density estimation and is a form of PCA, its expressivity is greatly limited, as evidenced by the authors' attempts to address these limitations using additional components like I-AE. Indeed, standard approaches often involve focusing solely on the diagonal elements or strictly enforcing the off-diagonals to be zero, as in [3]. However, these methods come with constrained expressivity.
>
> As detailed in the manuscript, the utilization of the L1 norm on the off-diagonal elements facilitates sparse learning and/or variable (non-strict) local orthogonality. This means that the chart's transformation can be significantly non-linear while retaining some of the benefits of orthogonal representations. Note that, the orthogonality does not mean a global coordinate system, only a local one. On the other hand, the L2-norm is used to match the metric to $\mathcal{I}$, ensuring an isometric embedding—an entirely distinct concept. As a result, we posit that our contribution is substantial not only as an independent method but also as a prototypical idea for optimization problems.
>
> W2: Indeed, a comparison with [1] should be incorporated into the relevant work section in the revised manuscript. Thank you for pointing this out. That said, given that our primary aim was to establish a highly expressive transformation devoid of heuristic solutions, which aligns with the core ideas of manifold learning flows, we focused on the most relevant part of the literature for brevity.
>
> Indeed, [2] represents a progression beyond M-flows, while RNFs directly compute the JtJ term that M-flows neglect, overcoming the aforementioned limitation. RNFs can be seen as a parallel approach to DNF [2] but with a more direct methodology, circumventing potential ambiguities stemming from heuristic techniques like density denoising [2]. [2] will also be cited in the revised version, thank you for pointing this out.
>
> In reference to [3], PCA flow relies on a lower-bound estimation of the probability density to bypass the JtJ calculation. This bound is tight when complete principal component flow, as defined by them, is achieved. The fundamental distinction here is that CMF does not confine itself to a pure PCA flow scenario, which has the potential to restrict expressivity. Since our method only loosely enforces orthogonality, or mutual information $\rightarrow 0$, we anticipate the contour lines to lie somewhere between NF and PCA flow. Furthermore, the introduction of sparsity adds an extra layer of complexity, making direct comparison challenging. As an outlook, we acknowledge that [2]  and possibly [3] see Q6, can undergo quantitative comparison with our method. However, at present, due to time constraints and the necessity for GPU cluster maintenance, we have been unable to carry out such comparisons.
>
> W3: It is accurate that any Riemannian manifold can be characterized by an orthogonal local basis. We merely use 4.1 to precisely define the term "canonical manifold", given that it is no a standard term in literature. We also include sparsity as part of the definition.
> Generally, the meaning of "canonical manifold" can vary depending on the context in which it is used. It is astutely pointed out that Definition 4.1 aligns closely with the assumption of PCA flow. However, it is vital to note that our method does not enforce this strictly. To put it in perspective, our approach seeks a 'partly canonical manifold,' if you will. Additionally, our method encompasses sparsity—meaning the diagonal elements can also approach zero.
>
> W4: Excellent remark, yes, we do employ the full dimension, D=d, for the low-dimensional simulated data, and this choice serves illustrative purposes. First, it allows us to visualize what all the latent dimensions are doing in relation to what we expect them to do. In particular, CMF correctly uses only 2 latent dimensions for representing 2D surfaces even though it has 3 latent dimensions. Second, it clearly shows the advantage to RNFs for the case where dimensions of the embedded manifold are known.  In principle, that can be a solution for higher-dim data. However, in practice it is very computationally expensive to train a full latent dimension (e.g. JtJ calculation) and the solution will take long to converge. Additionally, empirical knowledge suggests that a lower-dimensional representation can enhance expressivity, as discussed in the context of M-flows. Consequently, in practice, one can start from a predefined lower-dim and let the network optimize at will.
>
> W5: We apologize for any confusion cost. At Line 83 the ``$\theta$'' is a typo. $\mathcal{M}$ represents the lower dim manifold as learned by standard manifold learning, and $\mathfrak{M}$ the manifold learned by our method this convention is used to distinguish them, as done in lines 212 and 213. The data manifold is also a distinct manifold in that regard. %If such notation is confusing, we can simplify and just use words instead of the symbols.  We estimate that 4.1 is useful for the reasons elaborated above.
>
> Q1: 4.1 is used to define the canonical manifold (see W3).
>
> $*$Kindly refer to the global rebuttal for the continuation of the reply

---

> > ### Comment · Reviewer_iKjw · 2023-08-10
> > **Answer**
> >
> > I appreciate the detailed answers of the authors. I am mainly satisfied with them, and they help me to further appreciate the conceptual contribution of the paper - in particular the usage of the L1 error and the corresponding advantages. I will adjust my score accordingly.
> >
> > Regarding my questions:
> >
> > *Q2*: I'm afraid I disagree. The authors of "Rectangular Normalizing Flows" define them only for $d<D$. If $d=D$, an RNF should result in a standard NF. Please correct me if I am wrong. However, using a standard NF on a hollow sphere or Moebius band, Figure 2, should lead to very different results (the density must, in fact, degenerate unless you add some noise, which you did not mention).
> >
> > *Q3*: I don't quite see why this is computationally more expensive. The only computational impact will have the computation of the metric tensor. However, this only requires a backward pass which is efficient to compute. Please elaborate/correct me.
> >
> > *Q8:* This intuition should be part of the discussion in my opinion.
> >
> > *Q11: " For the analysis plots, we order the dimensions according to their weights and strictly choose a pre-defined number of them where applicable. "* So you need, in the end, an ID estimation. If so, this should be also clearly discussed in the limitations.

---

> > > ### Author Response · Authors · 2023-08-16
> > >
> > > We greatly appreciate the valuable insights offered in the review and the subsequent responses.
> > >
> > > Q2: Indeed, it is right that it would be equivalent to a standard NF. Moreover, as aptly noted, effectively learning these manifolds remains a challenge, even with the full-dimensional latent space. However, the RNF is based on CIF [5] which is an NF that already tries to solve the learning pathologies of complicated manifolds, "CIFs are not subject to the same topological limitations as normalising flows". Furthermore, in that regard, we estimate there is no practical limitation in setting $d=D$, as also seen empirically. Considering the above, the CMF method is showcased to outperform these previous methods, which also aligns with the theoretical intuition. We extend our appreciation to the reviewer for the thought-provoking discussion, and we'll certainly incorporate a comprehensive comment in the revised manuscript to reflect this exchange.
> > >
> > >
> > >
> > >
> > > Q3: The bottleneck for both methods is the calculation of the Jacobian-transpose-Jacobian (JTJ), as explained in Appendix 4 and RNF. The complexity of the approximation used is $\mathcal{O}(id^2) < \mathcal{O}(d^3) $ if $i << d$, $i$ refers to iterations of the iterative conjugate gradients method. M-Flow seeks to circumvent this computation entirely, while RNF aims for an approximation. Notably, as accurately noted, the JTJ yields the metric tensor directly, incurring no significant additional computational expense. When $d=D$, as previously discussed, this equates to a standard NF. In this regard, manifold flow methods were prompted to enhance computational efficiency by setting $d<<D$.
> > >
> > > Indeed, in alignment with the reviewer's observation, and as addressed here, the efficient approximate methods indicate that CMF could indeed be implemented for a full-dimensional flow. Further exploration of this potential could be undertaken in future studies. We will ensure to provide a diligent commentary in the revised manuscript.
> > >
> > > Q8 and Q11: We wish to clarify that this is solely necessary for Figure 4 and potentially for any future use that entails isolating prominent components. In this context, we sincerely appreciate the suggestion, which undoubtedly represents a substantial improvement over our current approach of analysis. Rest assured, we will incorporate these discussions into the updated version of the manuscript.
> > >
> > > [5] Relaxing Bijectivity Constraints with Continuously Indexed Normalising Flows
> > > Rob Cornish, Anthony L. Caterini, George Deligiannidis, Arnaud Doucet

---

> > > > ### Comment · Reviewer_iKjw · 2023-08-17
> > > > **Answer**
> > > >
> > > > Thank you for this clarification.

---

### Official Review · Reviewer_8w9Q · 2023-07-05

**Soundness:** 3 good
**Presentation:** 3 good
**Contribution:** 3 good
**Rating:** 5
**Confidence:** 3

**Summary:**

The paper studies the current manifold learning methods. It compares canonical manifold learning flow (CMF) with other manifold learning methods and demonstrates that CMF can learn the orthogonal features existing in data. From synthetic data on Moebius, the paper shows the benefits of using CMF.  Also, the paper shows the generated images from the CMF learned latent space is of higher quality than other methods' learned latent spaces. Lastly, the paper acknowledges some limitations of CMF, for example, high computation cost of the full Jacobian tranpose Jacobian.

**Strengths:**

1. The paper is very clear on the advantages of CMF against other manifold learning methods.
2. The paper did various types of experiments to showcase the CMF's performance among the methods.

**Weaknesses:**

1. Although being mentioned in the limitations, the paper does not provide results about the computation costs of CMF. There is no comparison of the computation time of CMF against other methods, also how the dimension affects the computation cost. It is unclear if CMF is scalable to higher-dimensional datasets.
2. The paper does not compare CMF with a lot of methods, including PCA and ICA or purely deep learning methods. For example, an autoencoder can also extract feature representations from data. How does CMF compare to these approaches?

**Questions:**

1. What is the complexity of the RNF and CMF methods?
2. For table 2, are there explanations on why M-flow works the best and CMF is the worst on GAS?
3. Figure 3 is hard to interpret. What are the advantages of CMF in this plot?
4. What are the generative models used in the image experiments?

**Limitations:**

The authors have discussed the limitations clearly in the paper. They acknowledge the high computation costs and the limitation that the method only applies when the manifold is homogeneous to Rd space.

---

> ### Author Rebuttal · Authors · 2023-08-09
>
> W1: We appreciate the reviewer for highlighting this matter. A comparative table can be found in Appendix 6 of the supplementary material. To summarize, there exists no substantial computational cost distinction between the RNF and CMF methods, as both encounter the calculation bottleneck of the Jacobian-transpose-Jacobian. Additionally, results from an initial CelebA 64x64 training have been included in the supplementary response PDF, indicating that CMF is scalable to higher-dim datasets with promising results. Currently,  manifold learning flows are computationally demanding; nevertheless, their potential is noteworthy due to their mathematical soundness, particularly in the case of methods such as RNF and CMF, which do not implement any fundamental approximations in the density calculations using lower dimensional latent space.
>
> W2: We believe this question is rather related to general manifold flow literature and not specific to our work. Linear PCA / ICA would fail when you apply them to data living on non-linear manifolds, for instance those shown in the toy examples. Non-linear PCA or non-linear ICA may be applied but one needs to identify the feature extractor or the kernel. Manifold learning methods come to aid at this point. They learn the non-linear transformation from the data. The method presented here falls in this category. Regarding comparing with AutoEncoders, AE's would not learn data distributions, e.g., they cannot be used to sample or compute likelihoods. Manifold flow learning techniques, to which our method belongs to, fall in the category of methods that approximate distributions. Furthermore, manifold learning flow methods are relatively new and in that regard there are only a couple of prominent works.
>
> Q1: The bottleneck for both methods is the calculation of the Jacobian-transpose-Jacobian, as explained in Appendix 4 and [1]. The complexity of the approximation used is $\mathcal{O}(id^2) < \mathcal{O}(d^3) $ if $i << d$, $i$ refers to iterations of the iterative conjugate gradients method and d to the latent dimension.
>
> Q2: We appreciate the reviewer's comment. As indicated in the manuscript, we calculate the average best FID score from an average of 5 simulations for each tabular dataset.  We  have conducted an additional GAS tabular experiment and present the results in an updated FID score table for tabular datasets, accessible in the supplementary PDF. Notably, the performance ordering has shifted, indicating possible inherent challenges in learning the dataset, which might introduce randomness in the outcome. However, it's worth mentioning that all other tabular datasets maintained consistent performance orderings across methods, so we haven't included them here.
>
> To delve deeper into this matter, we visualize the validation FID-like scores during training, available in the supplementary PDF. Upon comparing the results of Hempmass and GAS, it's apparent that learning converges until a certain point, after which instability may arise for all three methods. Despite this instability, it's important to note that our implementation selects the best validation step checkpoint for testing, mitigating the impact of training instability. However, it's plausible that these instabilities contribute to the variations between different runs. Furthermore, we have introduced another tabular dataset, namely Miniboone,  as well as we run experiments with double the latent dimensions as compared to the original experiments, all with promising results.
>
>
> Q3: Figure 3 demonstrates the realization of sparse and partially orthogonal learning, as per hypothesis of the proposed method, when compared with the original approach. Additionally, when these findings are coupled with the better FID scores for the image datasets and the outcomes from the simulated data, it can be deduced that CMF is opting for a more ``efficient'' use of the latent space, in other words enforcing orthogonality between dimensions and not putting any weight on unnecessary dimensions. We acknowledge that visualizing high-dimensional data always presents a challenge.
>
>
> Q4: We wish to clarify that the CMF framework functions as an independent generative model. It approximates the data distribution with a low dimensional latent space. The learned manifold pertains to the lower-dimensional representation of the data, which is subsequently mapped back into the image space, allowing direct sampling from the latent space for image generation. The essence of this work is to demonstrate that by acquiring a more effective latent representation, we achieve a more successful generative model than its precursors, M-flow and RNF!
>
> [1] Rectangular Flows for Manifold Learning Anthony L. Caterini, Gabriel Loaiza-Ganem, Geoff Pleiss, John P. Cunningham

---

> > ### Comment · Reviewer_8w9Q · 2023-08-19
> >
> > Thank you for addressing my questions and concerns. I am more leaning to accept now and raised the score to 5.

---

> > > ### Author Response · Authors · 2023-08-20
> > >
> > > We would like to express our gratitude to the reviewer for their evaluation.

---

### Official Review · Reviewer_7TiS · 2023-07-07

**Soundness:** 4 excellent
**Presentation:** 4 excellent
**Contribution:** 3 good
**Rating:** 7
**Confidence:** 4

**Summary:**

The authors introduce a new method for regularising manifold learning flows. Essentially, it attempts to reduce the entanglement between the dimensions of the learned manifold by encouraging non-diagonal elements of the metric tensor to be small. Leveraging the already necessary computation of $J^\top J$, where $J$ is the Jacobian of the flow transformation, this is done efficiently by minimising the $\ell-1$ norm of the non-diagonal entries.

Experiments on synthetic and real data show how this is effectively achieved, especially when compared to similar models that do not employ the regularisation scheme. Beyond achieving the desired effect, the proposed method achieves lower FID-like scores on real tabular data.

**Strengths:**

The paper is very well written and the presented ideas are easy to follow. Experimentally, the authors confirm both in intuitive, synthetic examples and real data. The proposed regularisation scheme is also simple and computationally efficient, which is also a desirable feature. Overall, I consider it a strong contribution and see high potential of being widely adopted as a mechanism for regularising manifold learning flows.

I also appreciate the limitations raised and discussed by the authors, indicating maturity in their analysis and raising important considerations related to the use of their work.

**Weaknesses:**

Although I see the potential mentioned above, I do consider it could be viewed more widely as a weak point, given the relatively niche application. Although it does not affect the merit of the work, the principle might be too specific to manifold learning flows.

I would consider more tabular data experiments are needed, with additional data sets, given the mixed results attained. More concretely, some analysis of what specifically differentiates the data sets enough to cause the difference in performance.

**Questions:**

- Tied to the aforementioned weakness, what do the authors posit is the cause for the performance gap in GAS?
- Related to the previous question: are the metric tensors found in tabular data also behaving similarly to the ones shown in Figure 3?
- Is it clear during training when a pathological scenario is reached? How does it behave w.r.t. increasing/decreasing the value of $d$?
- Are the learned manifold representations also useful as a feature extraction procedure for downstream tasks? Could this be also a scenario for evaluating the quality of the learned models? (As opposed to only raw performance in scores)

Minor comments:
- line 35: Caterini et al [7] has => Caterini et al. [7] have
- line 44: Please avoid unnecessary adjectives such as "tedious"
- line 62: why is the set starting with $x_0$?

**Limitations:**

I was pleasantly surprised with the discussion on limitations in the paper. I believe the authors addressed important issues and even highlighted which other approaches could have better raw performance.

---

> ### Author Rebuttal · Authors · 2023-08-09
>
> We extend our gratitude to the reviewer for their precise summary.
>
> W1: We thank the reviewer for the insightful remarks. Indeed, we have developed the method in the context of manifold learning flows in order to solve an existing pathology as well as dis-entagling the latent space. While being specific at the moment, it is our estimate that there is a prototypical idea and thus of some theoretical value as well. Specifically, we propose and show that the off-diagonal manifold metric elements when minimised by an L1 loss allow simultaneous sparse and/or orthogonal basis. This idea can be used in other optimization schemes.
>
> W2: We thank the reviewer for the suggestion. We carried out one more tabular data experiment, experiments with twice the size of latent dimensions and repetitions of the experiments for understanding. We show the results in the supplementary PDF. See also Q1 for discussion on these and their training curves.
>
> Q1: We appreciate the reviewer's suggestion. As indicated in the manuscript, we calculate the average best FID score from an average of 5 simulations for each tabular dataset. In response to your suggestion, we conducted an additional GAS tabular experiment and present the results in an updated FID score table for tabular datasets, accessible in the supplementary PDF. Notably, the performance ordering has shifted, indicating possible inherent challenges in learning the dataset, which might introduce randomness in the outcome. However, it's worth mentioning that all other tabular datasets maintained consistent performance orderings across methods, so we haven't included them here.
>
> To delve deeper into this matter, we visualize the validation FID-like scores during training, available in the supplementary PDF. Upon comparing the results of Hempmass and GAS, it's apparent that learning converges until a certain point, after which instability may arise for all three methods. Despite this instability, it's important to note that our implementation selects the best validation step checkpoint for testing, mitigating the impact of training instability. However, it's plausible that these instabilities contribute to the variations between different runs. Furthermore, following your suggestion, we have introduced another tabular dataset, namely Miniboone.
>
> Q2: Indeed, they behave similarly. We visualize these  for the tabular datasets trained with sufficiently high latent dimensions and present the mean absolute cosine similarity (MACS) in the supplementary PDF. For the specific case of GAS with d=2, the plot might not offer significant insights. However, we  report here the MACS for RNF as $3.16 \pm 0.7$ and for CMF as $2.16 \pm 0.2$. We observe that the MACS is lower for CMF, although, considering the error, there remains some overlap, which is within expectations.
>
> Q3: We conducted training for the tabular datasets with twice the number of their respective latent dimensions, as compared to the original,  see Appendix. The outcomes are available in the provided PDF. Notably, we haven't observed any substantial differences. Furthermore, the training curves follow similar behaviours, converging up to a certain point and then displaying a degree of instability. This trend holds true for all three methods when trained on the GAS dataset.
>
> Q4: That is part of the main outlook of this work. For example, improved out-of-distribution detection could be a consequence of the feature extraction, some preliminary but promising results have been included in the supplementary material of this work. Additionally, data that require orthogonal basis vectors, like solutions of a many-body-physics quantum Hamiltonian, can show improved learning performance with the current method.
>
> We acknowledge the minor corrections and will incorporate them into the revised version of the manuscript. The reviewer is correct, the typo $x_0$ should be rectified, and the set should commence with $x_1$ for consistency.

---

> > ### Comment · Reviewer_7TiS · 2023-08-20
> >
> > I thank the authors for addressing all issues I raised. After reading the other reviews and rebuttals, I am increasing my score to 7 (Accept), as I no longer think there are any outstanding issues with the submission.

---

> > > ### Author Response · Authors · 2023-08-20
> > >
> > > We would like to thank the reviewer for their valuable comments.

---

### Official Review · Reviewer_tk9x · 2023-07-08

**Soundness:** 2 fair
**Presentation:** 1 poor
**Contribution:** 2 fair
**Rating:** 5
**Confidence:** 4

**Summary:**

This paper studies the problem of learning a latent representation for data supported on a low-dimensional manifold. It proposes to promote orthogonality of the tangent vectors arising from a learned chart, on top of existing rectangular flow loss. Experiments are provided to demonstrate the effectiveness of the algorithm.

**Strengths:**

Promoting orthogonality of the tangent vectors from a learned chart is interesting.

**Weaknesses:**

## Presentation:
1. The paper seems insufficiently prepared and proofread. In particular, there are numerous evident typos even in the first paragraph of the introduction. Consequently, they weaken the credibility of the paper.
    1. Lines 29, 33 the "D" in "R^D" uses a mathbb, whereas in lines 31, 32 the "d" or "D" does not.
    1. Line 29 “fulfil” -> “fulfill”
    1. In equation 1, the q_phi(x) should have been q_phi^{-1} (x)
    1. Section 3 talks about general manifold learning, and is almost detached from the technical sections 2 and 4.
2. Line 43: A line is one-dimensional. What is a two-dimension line?
3. If one compares equations (5) and (6), it appears that we assume the learned map q_phi is a chart of manifold M. If that is the case, this should be made explicit.
4. From (8), it is unclear that where the introduced G term are evaluated. I suppose it is on points x_i’s?

**Questions:**

## Significance
Looks like the inverse of g_phi is in objective (6), so using line 75 it boils down to the inverses of g_eta and f_theta. How easy is it to invert g_eta and f_theta? The root of this question is, if one takes f_theta to be a square matrix, then inverting the square matrix is needed, leading to high cost if the dimension of the square matrix is large. If f_theta is a square matrix followed by say ReLU, then ReLU is not invertible.

**Limitations:**

Yes

---

> ### Author Rebuttal · Authors · 2023-08-09
>
> We appreciate the reviewer's suggestions. While we acknowledge that different presentation approaches could have been explored, we believed that, given space constraints, the current explanation best encapsulated the work.  We are encouraged by the fact that the presentation is also accepted by the other reviewers. We apologize for any typos or language errors and commit to rectifying them diligently in the revised paper.
>
>  1.1 Indeed, there was bracket typo in latex there.
>
>  1.2 In British English it is indeed "fulfil", but of course the convention is American English, we will fix this.
>
>  1.3 Apologies, for the typo.
>
>  1.4 Our method is based on manifold learning flows; consequently, the first paragraph is dedicated to that. Furthermore, the rest of the section, and indeed the majority of it, addresses the relevant work that inspires our method or explores similar themes. This includes the Relevance Vector Machine, which motivates sparse learning, PCA and ICA methods that emphasize orthogonality, tabular neural networks which implement a similar idea to ours to constrain a specific NN, and PCA flow which explores orthogonal contour learning for flows.
>
>  2. Two-dimension here refers to the length and width of this 'fuzzy line'. Indeed, a line is a 1D object but the "fuzzy line" we use here has 2D noise added to the line.
>
>  3. Line 172, just above equation 6 we explicitly specify 'the transformation of the chart'
>
>  4. This is correct, G is depended on x and the parameters of the network, we understand this is explicitly defined by equations (6) and (7), of course we can include this in G or explicitly specify that the dependencies are dropped for brevity.
>
>
>  This is indeed a fundamental property of invertible flows, and it has ultimately been tackled through various methods, such as 'coupling layers' [1], in the last 5-10 years. Additionally, the inconsistency with the activation function is an acknowledged limitation of such approaches that has already been addressed over these years [2]. For instance, when the ReLU activation is utilized in a normalizing flow, it is typically combined with an element-wise scale and shift transformation (affine transformation) to ensure invertibility. The original flow part of our work builds upon these prior contributions and more recent developments [3].
>
> [1] Laurent Dinh, David Krueger, Yoshua Bengio NICE: Non-linear Independent Components Estimation
>
> [2] Real NVP: Density Estimation and Inverse Problems Laurent Dinh, Jascha Sohl-Dickstein, Samy Bengio
>
> [3] Variational Inference with Continuously-Indexed Normalizing Flows Anthony Caterini, Rob Cornish, Dino Sejdinovic, Arnaud Doucet

---

> > ### Comment · Reviewer_tk9x · 2023-08-20
> >
> > Thank you for the response.
> >
> > 1.2: I apologize for being ignorant of the spelling - British English, as any other English, is perfectly fine.
> >
> > 1.1-1.3: My intention was not to be nit-picking. Everyone makes typos and mistakes, and that is what proofreading is for. Having obvious typos in early parts of the paper is a sign of a lack of proofreading, thus it lowers the credibility of other parts that are harder to check.
> >
> > Nonetheless, after reading the response and other reviews, I raised my rating.

---

> > > ### Author Response · Authors · 2023-08-20
> > >
> > > We extend our appreciation to the reviewer for offering clarifications and investing the time to re-examine our work.

---

### Author Rebuttal · Authors · 2023-08-09

We express our gratitude to both the reviewer and the chair for their valuable time and insights. We have diligently addressed each of the reviewer's comments individually. Furthermore, we have expanded our testing to encompass additional tabular datasets and incorporated CelebA, 64x64 FID test scores for a more comprehensive evaluation.

$........................................................................................$

In order to adequately address the points raised in the extended review by reviewer 4 (8w9Q), we make use of this space for further discussion.

Q2: The comparison is indeed fair as they are both manifold learning flows, nothing restricting the use of $d=D$. Furthermore, similar results can be obtained by embedding a 2-sphere in a 4D space and setting $d=3$, or just the orthogonality can be see by setting $d=2$. Additionally, the arguments about lower-dim expressivity are not so relevant for such low-dim data, see W4.

Q3: Excellent remark, it is mathematically sound but computationally expensive and possible convergence issues, see W4.

Q4: Yes, exactly for this reason, CMF is shown to be learning better. Perfect density estimation is not a trivial task from our experience.

Q5: The log likelihood (L)  is 1.6553 and 1.6517, and the KS p-value is 0.17 and 0.26, for the CMF and RNF methods respectively. For the sphere,  L=1.97 and 1.16 for for the CMF and RNF methods respectively. CMF shows improved quantitative quality over the RNF, as it is also seen qualitatively.

Q6: PCF [3], mentions that they obtain a similar test set log likelihood to that of the normalizing flow (NF), and that the PCF has almost zero pointwise mutual information. Additionally, the M-Flow method shows already improved image generations over other flow methods, and RNF improves on those even further.  Therefore, we had estimated that RNF is the main competitor in this regard. Furthermore, there can be no real comparison of the mutual information (or similarity etc) as the PCF relies on fully orthogonal contours, which is not necessarily a desirable quality in the CMF framework as explained in W1. Nevertheless, a comparison can always be made.  However, due to time constraints and GPU cluster maintenance, it is currently pending.

Q7: We appreciate you bringing this to our attention. Our methodology was to run about 10 repetitions for each and chose the best. Our estimate is that it is a hard manifold with non-trivial topology that degenerate latent-space representations fail to capture it well.

Q8: An excellent question, it is more an intuition that the largest eigenvalues (if an eigen-decomposition is possible) correspond to the dimensions with the highest weights.

Q9: Enforcing strict complete orthogonality can restrict the expressivity of the transformation in the case of high dimensional complex data where the latent manifold is completely unknown.

Q10: L2 with the identity matrix can be used to enforce a specific embedding, e.g. isometric [1] or conformal [4] which is not the purpose of this work, we want the learned transformation to be general as explained in W1 as well.

Q11: We appreciate you bringing this to our attention and allows us to clarify that the method does not arbitrarily discard any dimensions, rather they would be effectively irrelevant if their weight is close to zero. For the analysis plots, we order the dimensions according to their weights and strictly choose a pre-defined number of them where applicable. Thresholds would be very hard to define indeed.

Minor: We acknowledge, and we will diligently rectify, all the minor issues in the revised manuscript. Notes: line 83: $\theta$ was a typo. line 225: We mean that the method is not overly sensitive to the hyperparameter $\gamma$, we will rephrase, thank you. line 299: This is the main premise of RNF and Denoising Normalizing flows, also this limitation is accepted in the original M-Flow paper.

Lim: As elaborated in Q11, there is no need to find the prominent latent dimensions specifically, their weights define their significance as adjusted by sparse learning and/or orthogonality.

[4] Tractable Density Estimation on Learned Manifolds with Conformal Embedding Flows
Brendan Leigh Ross, Jesse C. Cresswell

---

> ### Author Response · Authors · 2023-08-20
>
> We sincerely appreciate the time and effort that all the reviewers have dedicated to assessing our work, and we express our gratitude to them.

---

### Decision · Program_Chairs · 2023-09-21

**Decision:**

Accept (poster)

**Comment:**

After the rebuttal and some discussion the reviewers agree that the paper is suitable for acceptance.  A number of small issues have been identified during the review process and the authors should revise their manuscript accordingly for the final version.